# Improving age-depth relationships by using the LANDO model ensemble

Gregor Pfalz[1,2,3,4], Bernhard Diekmann[1,2], Johann-Christoph Freytag[3,4], Liudmila Syrykh[5], Dmitry A. Subetto[5,6], Boris K. Biskaborn[1,2]

[1]Alfred Wegener Institute, Helmholtz Centre for Polar and Marine Research, Research Unit Potsdam, Telegrafenberg A45, 14473 Potsdam, Germany
[2]University of Potsdam, Institute of Geosciences, Karl-Liebknecht-Str. 24-25, 14476 Potsdam-Golm, Germany
[3]Einstein Center Digital Future, Robert-Koch-Forum, Wilhelmstraße 67, 10117 Berlin, Germany
[4]Humboldt-Universität zu Berlin, Department of Computer Science, Unter den Linden 6, 10099 Berlin, Germany
[5]Herzen State Pedagogical University of Russia, Moyka Emb. 48, St. Petersburg 191186, Russia
[6]Institute for Water and Environmental Problems of the Siberian Branch of the Russian Academy of Sciences, Molodezhnayastr.1, Barnaul 656038, Russia

*Correspondence to:* Gregor Pfalz (Gregor.Pfalz@awi.de), Boris K. Biskaborn (Boris.Biskaborn@awi.de)

## Abstract

Age-depth relationships are the key elements in paleoenvironmental studies to place proxy measurements into a temporal context. However, potential influencing factors of the available radiocarbon data and the associated modeling process can cause serious divergences of age-depth relationships from true chronologies, which is particularly challenging for paleolimnological studies in Arctic regions. This paper provides geoscientists with a tool-assisted approach to compare outputs from age-depth modeling systems and to strengthen the robustness of age-depth relationships. We primarily focused on the development of age determination data from a data collection of high latitude lake systems (50° N to 90° N, 55 sediment cores, and a total of 602 dating points). Our approach used five age-depth modeling systems (*Bacon, Bchron, clam, hamstr, Undatable*) that we linked through a multi-language Jupyter Notebook called **LANDO** ("**L**inked **a**ge a**n**d **d**epth m**o**deling"). Within LANDO we have implemented a pipeline from data integration to model comparison to allow users to investigate the outputs of the modeling systems. In this paper, we focused on highlighting three different case studies: comparing multiple modeling systems for one sediment core with a continuously deposited succession of dating points (CS1), for one sediment core with scattered dating points (CS2), and for multiple sediment cores (CS3). For the first case study (CS1), we showed how we facilitate the output data from all modeling systems to create an ensemble age-depth model. In the special case of scattered dating points (CS2), we introduced an adapted method that uses independent proxy data to assess the performance of each modeling system in representing lithological changes. Based on this evaluation, we reproduced the characteristics of an existing age-depth model (Lake Ilirney, EN18208) without removing age determination data. For multiple sediment cores (CS3) we found that when considering the Pleistocene-Holocene transition, the main regime changes in sedimentation rates do not occur synchronously for all lakes. We linked this behavior to the uncertainty within the dating and modeling process, as well as the local variability in catchment settings affecting the accumulation rates of the sediment cores within the collection near the glacial-interglacial transition.

## 1 Introduction

Lakes sediments are important terrestrial archives for recording climate variability in the high latitudes of the Northern Hemisphere (Biskaborn et al., 2016; Smol, 2016; Lehnherr et al., 2018; Subetto et al., 2017; Syrykh et

al., 2021; Diekmann et al., 2017). The identification of age-depth relationships in those lake sediments helps us to put their measured sediment properties in a temporal context (Bradley, 2015; Lowe and Walker, 2014; Blaauw and Heegaard, 2012). We can determine these relationships by directly counting the annual laminated layers (varves) (Brauer, 2004; Zolitschka et al., 2015), or by using indirect age determination methods such as radiocarbon, optically stimulated luminescence (OSL), or lead-cesium (Lead-210/Cesium-137) dating (Lowe and Walker, 2014; Bradley, 2015; Appleby, 2008; Hajdas et al., 2021). Defining a reliable age-depth relationship for paleoenvironmental studies in cold regions is particularly challenging, as varves only exist in rare cases and the determination of ages mostly depends on radiocarbon dating (Strunk et al., 2020 and references therein). Because of primarily financial restrictions, however, only a few selected samples are taken from sediment core sections to determine the corresponding ages of certain depths (Blaauw et al., 2018; Ciarletta et al., 2019; Olsen et al., 2017). We therefore rely on model calculations to define the ages between the samples. In addition to the mathematical challenges that arise when establishing age-depth relationships, the selection of appropriate dating material has an impact on the modeling process.

In the special case of Arctic lake systems, the amount of material for radiocarbon dating, i.e. aquatic/terrestrial macrofossils and organic remains, is extremely low (Abbott and Stafford, 1996; Colman et al., 1996; Strunk et al., 2020). Radiocarbon dating is therefore often based on the organic carbon content in bulk sediment samples, which can be relatively small due to the lower bioproductivity in those lakes (Strunk et al., 2020 and references therein). However, the use of bulk sediments is problematic, as some portions of contributing carbon are not occurring at the same time as the deposition but may reveal inherited ages from reworked older materials (Rudaya et al., 2016; Biskaborn et al., 2013b, 2019; Schleusner et al., 2015). Several methods are available for pre-treating bulk sediment samples to address sample-based dating uncertainties (Brock et al., 2010; Strunk et al., 2020; Rethemeyer et al., 2019; Bao et al., 2019; Dee et al., 2020). Each pre-treatment method may yield a different result for the same material due to the influence of humic acids, fulvic acids, and humins (Brock et al., 2010; Strunk et al., 2020; Abbott and Stafford, 1996). Similarly, older, inert material incorporated by living organism, known as "reservoir effect" or "hard-water effect", distorts the actual radiocarbon age by up to ±10 000 years (Ascough et al., 2005; Austin et al., 1995; Lougheed et al., 2016). Such a distortion creates methodological and mathematical errors in the development of age-depth relationships, which possibly leads to a misinterpretation of these relationships.

There are numerous geochronological software systems (from now on simply called modeling systems) available to the geoscientific community, which try to solve the challenges stated above (Trachsel and Telford, 2017; Wright et al., 2017; Lacourse and Gajewski, 2020). Implemented methods for detecting outliers, accounting for varying sedimentation rates, or using bootstrapping processes support the construction of an age-depth model (Parnell et al., 2011; Lougheed and Obrochta, 2019; Bronk Ramsey, 2009, 2008).

However, the correct usage of those systems requires a high degree of understanding of the underlying mathematical methods and models. Trachsel and Telford (2017) noted that, despite the users' impact on the outcome of the model by setting priors and parameters, most users do not have any prior objective insights into appropriately choosing the right parameters. Wright et al. (2017), Trachsel and Telford (2017), and Lacourse and Gajewski (2020) even showed that the results produced by modeling systems could diverge from the true chronology. An in-depth comparison of the results is therefore extremely error-prone. Due to time constraints, usually, users only select and apply one modeling system for paleoenvironmental interpretation.

The objective of this paper is to reduce the effort involved in applying different methods for determining age-depth relationships and to make their results comparable. We provide a tool to link five selected modeling systems in a single multi-language Jupyter Notebook. We introduce an ensemble age-depth model that uses uninformed models to create data-driven, semi-informed age-depth relationships. We demonstrate the power of our tool by highlighting three case studies in which we examine our application for individual sediment cores and a collection of multiple sediment cores. Throughout this paper, the term "**LANDO**" refers to our implementation, which stands for "**L**inked **a**ge a**n**d **d**epth m**o**deling". The current development version of LANDO is accessible via GitHub (https://github.com/GPawi/LANDO).

In this paper, we use published age determination data from 55 sediment cores from high latitude lake systems (50° N to 90° N). This unique collection of age determination data allows us to thoroughly test LANDO by examining changes of sedimentation rates over time for various modeling and lake systems. The harmonization of the acquired data follows the conceptual framework described in Pfalz et al. (2021).

## 2   Methods

A key element in our data-science based approach for developing comparable age-depth relationships was to facilitate the use of modeling systems independent from their original proprietary development environment. A multi-language data analysis environment, such as the SoS notebook (Peng et al., 2018) or GraalVM (Niephaus et al., 2019), provides an interface that enables the comparison of modeling systems without being limited to one programming language or environment. Our implementation used the SoS notebook as its backbone. The SoS notebook is a native Python- and JavaScript-based Jupyter Notebook (Kluyver et al., 2016), which extends to other languages through so-called "Jupyter kernels". We developed our implementation with the focus on four languages and their respective kernels: Python, R, Octave, and MATLAB. This selection allowed us to use the most common modeling systems.

According to Lacourse and Gajewski (2020), the most commonly used modeling systems are *Bacon* (Blaauw and Christen, 2011), *Bchron* (Haslett and Parnell, 2008; Parnell et al., 2008), *OxCal* (Bronk Ramsey, 1995; Bronk Ramsey and Lee, 2013), and *clam* (Blaauw, 2010). We additionally considered the MATLAB/Octave software *Undatable* (Lougheed and Obrochta, 2019), as an alternative to the classical Bayesian approach, and the R package *hamstr* (Dolman, 2022).

In our study, we were able to connect five of the above-mentioned modeling systems in the SoS notebook, namely: *Bacon, Bchron, clam, hamstr*, and *Undatable*. All modeling systems assume a monotonic deposition process, i.e. a positive accumulation rate over the entire core length (Trachsel and Telford, 2017; Lougheed and Obrochta, 2019). Modeling system *clam* uses five different regression-based techniques in combination with a Monte Carlo procedure to repeatedly interpolate between calibrated dates. Because *clam* tries to fit the regression curves to the data, in some cases this can lead to age inversions, which *clam* automatically filters out. (cf. Trachsel and Telford, 2017; Blaauw, 2010)

The modeling procedure of *Undatable* involves a weighted random sampling from both calibrated age and depth uncertainties (expressed as probability density functions) for all dating points and an advanced bootstrapping process over a user-defined number of simulations. The advanced bootstrapping procedure includes removing age

inversions from the simulation runs as well as inserting connection points between calibrated dates to account for uncertainties in sediment accumulation rates between the dating points. (cf. Lougheed and Obrochta, 2019)

The Bayesian modeling systems *Bacon*, *Bchron*, and *hamstr* subdivide the sediment core into smaller increments for the modeling process but differ in their division technique. *Bacon* separates the core into equal segments, while *hamstr* extends *Bacon*'s algorithm by adding additional hierarchical accumulation structures to each segment (Trachsel and Telford, 2017; Dolman, 2022; Blaauw and Christen, 2011). *Bchron* estimates the number of increments between calibrated dates by a compound Poisson-gamma distribution (Trachsel and Telford, 2017;

Parnell et al., 2011). For age-depth calculations, *Bacon* uses prior distributions for the accumulation rate (gamma distribution) and autocorrelation memory (beta distribution) between segments, which users can fit with values for the mean and shape of these distributions (Blaauw and Christen, 2011). Similarly, *hamstr* relies on user input for the shape of the gamma distribution and values for the memory but estimates the mean value for the accumulation rate from the available age determination data by using a robust linear regression (Dolman, 2022). *Bchron* does

not require any specific hyperparameters selection due to its fully automated numerical best-fit approach (Wright et al., 2017; Haslett and Parnell, 2008). All three Bayesian modeling systems use iterations of the Markov chain Monte Carlo (MCMC) algorithm to estimate the calibrated ages and confidence intervals at each depth within the sediment core (Dolman, 2022; Blaauw and Christen, 2011; Haslett and Parnell, 2008).

The workflow of LANDO consists of five major components: Input – Preparation – Execution – Result aggregation

– Evaluation of model performance.

## 2.1  Input

To work with LANDO users need to provide age determination data, e.g., data from radiocarbon or OSL dating, and associated metadata as listed in Table 1. We developed two import options for the users: through a single spreadsheet or a connection to a database. For this study, we used a connection to a PostgreSQL database, which

we developed after the conceptual framework as described in Pfalz et al. (2021), via the Python package "SQLAlchemy" (Bayer, 2012). We divided age determination input data into two attribute categories: necessary and recommended. The category "necessary" focused on the prerequisites of the individual modeling systems as well as project-related attributes, such as unique identifiers, i.e., "measurementid", "labid". However, a larger comprehensive set of descriptive metadata helps a better understanding of the data (Cadena-Vela et al., 2020;

Thanos, 2017). We added four additional attributes from the category "recommended" to facilitate the interpretation of age-depth models regarding their age determination data.

**Table 1** – *Necessary and recommended attributes for age determination input data, when used with LANDO. Attributes apply for both input methods through either a database or a spreadsheet.*

| Attribute | Description | Data type | Necessary/ Recommended |
|---|---|---|---|
| *measurementid* | Composite key composed of a unique CoreID, a blank space, and the depth below sediment surface (mid-point cm) with max. two decimal digits of corresponding analytical age measurement - *example: "CoreA1 100.5", when users obtained sample of CoreA1 between 100 and 101 cm depth* | string | Necessary |

| | | | |
|---|---|---|---|
| *thickness* | Thickness of the sample slice used for age determination in [cm] | float | Necessary |
| *labid* | Unique sample identifier that was provided by the laboratory for age determination | string | Necessary |
| *lab_location* | Name of city, where laboratory that conducted the analysis resides | string | Recommended |
| *material_category* | One of the eight categories that describes the material best, based on the categories from age-depth modeling system *Undatable* (Lougheed and Obrochta, 2019)<br><br>*14C marine fossil*   *tiepoint*<br>*14C terrestrial fossil*  *paleomag*<br>*14C sediment*     *U/Th*<br>*tephra*       *other* | string | Necessary |
| *material_description* | Short description of the used material | string | Recommended |
| *material_weight* | Weight of analyzed carbon used in radiocarbon dating in [µgC] | float | Recommended |
| *age* | Uncalibrated radiocarbon age in [uncal yr BP], or non-radiocarbon ages as values in [yr BP] (BP = Before Present (before 1950 CE)) | float | Necessary |
| *age_error* | Error of the uncalibrated radiocarbon age and non-radiocarbon age in [yr] | float | Necessary |
| *pretreatment_dating* | Concise description or abbreviation of sample pre-treatment - *example: "ABA", when radiocarbon pre-treatment comprises of an acid-base-acid sequence* | string | Recommended |
| *reservoir_age* | Additional reservoir effect (also known as hard-water effect or age offset) identified by the user in [yr]; if unknown, then insert 0 | float | Necessary |
| *reservoir_error* | Error of reservoir age known to the user in [yr]; if unknown, then insert 0 | float | Necessary |

If users decide to use a spreadsheet as input option, then the spreadsheet should follow the same attribution as the database. In addition, we implemented an input prompt for further information, such as the year of core drilling and core length, to ensure comparability to our database implementation. We provide an example spreadsheet with all attributes in the expected format in the repository mentioned in the "Code and data availability" section of this paper.

**2.2 Preparation**

The preparation component consisted of two separate steps. First, we checked each age determination dataset, whether a reservoir effect was influencing the radiocarbon data. In the absence of a known reservoir age or recent surface sample, we used available radiocarbon data points and a fast-calculating modeling system to predict the age of the upper most layer within a sediment core. In our approach, we used the *hamstr* package with a default

value of 6000 iterations. We then compared the predicted value for the upper most layer with the year of the core retrieval, i.e., our target age. We accounted for an uncertainty in the estimate by allowing an extra 10% error between predicted age and target age. If a gap between predicted and target age is observable, then we assumed a reservoir effect is present. We calculated the reservoir effect by subtracting the target age from the mean predicted age, whereas the associated error we based on the two-sigma uncertainty ranges of the prediction. LANDO allow

users to add the calculated reservoir age and its uncertainty range to the corresponding attributes ("reservoir_age" and "reservoir_error"). Depending on the choice of the user, this addition affects either all radiocarbon samples or only bulk sediment samples, or users completely discard the output for the subsequent modeling process.

As second step in the preparation component, we built a module that automatically changes the format of the available data to the individually desired input of each of the five modeling systems implemented in LANDO. We

primarily used the Python package "pandas" (Reback et al., 2020) for the transformation within the module. We transferred the newly transformed age determination data to the corresponding programming language for age-depth modeling using the built-in "%get" function of SoS notebook.

### 2.3 Execution

We developed LANDO with the specific ability of create multiple age-depth models for multiple dating series

from spatially distributed lake systems. Hence, reducing overall computing time was one of our highest priorities. We achieved this reduction by applying existing parallelization back-ends for both R and Python, such as "doParallel" (Microsoft Corporation and Weston, 2020a) and "Dask" (Dask Development Team, 2016), respectively. For each modeling system in R, we wrote a separate script that takes advantage of the parallelization back-end "doParallel". Besides the individual modeling system packages, we made use of different R libraries,

such as "tidyverse" (Wickham et al., 2019), "parallel" (R Core Team, 2021), "foreach" (Microsoft Corporation and Weston, 2020c), "doRNG" (Gaujoux, 2020), and "doSNOW" (Microsoft Corporation and Weston, 2020b). We neglected the use of parallelization for the *Undatable* software in MATLAB, since even the sequential execution for several sediment cores in our test setup was on the order of a few minutes. However, we achieved comparable results with *Undatable* in Octave using the parallelization package "parallel" (Fujiwara et al., 2021).

As mentioned before, the selection of model priors and parameters has an impact on the modeling outcome, if no objective prior knowledge exist. To lower our impact and to avoid introducing biases in the modeling process, we used the default values from each modeling system as our own default values (Blaauw et al., 2021; Blaauw, 2021; Parnell et al., 2008; Dolman, 2022; Lougheed and Obrochta, 2019). In our adaptation of clam, the parameter "poly_degree" controls the polynomial degree of models for type 2, while the parameter "smoothing" controls the

degree of smoothing for type 4 and 5. In the original version of clam, users adjust both parameters with the single option "smooth" (Blaauw, 2021). Furthermore, the default value for "ssize" within the original version of Bacon is 2000. We increased this value to 8000 to ensure good MCMC mixing for problematic cores (Blaauw et al., 2021). In case the user has in-depth knowledge about his sediment core and wants to change certain values, we opted for making crucial parameters accessible within the SoS notebook outside of the executing scripts. Table 2

provides an overview of all values which users can access and change for the individual systems. However, we limited the access to some parameters for operational purposes, such as the number of iterations or the resolution of the output.

**Table 2** – *Default values for each modeling system, which users can access and change within LANDO.*

| Modeling system | Parameter | Default value |
| --- | --- | --- |
| *Bacon* | | |
| | acc.shape | 1.5 |
| | acc.mean | 20 |
| | mem.strength | 10 |

| | | |
|---|---|---|
| | mem.mean | 0.5 |
| | ssize | 8000 |
| *Bchron* | *not applicable* | - |
| *clam* | types | 1 to 5 |
| | poly_degree | 1 to 4 |
| | smoothing | 0.1 to 1.0 |
| *hamstr* | K | c(10,10) |
| *Undatable* | xfactor | 0.1 |
| | bootpc | 30 |

## 2.4 Result aggregation

After every model run, we received 10000 age estimates (also known as "iterations" or "realizations") per centimeter from each modeling system for every sediment core. We transferred these results back to Python using the built-in "%put" function of SoS notebook, where in the next module, we calculated per centimeter the median and mean age values as well as one-sigma and two-sigma age ranges. For the summarizing statistics, we used standard Python libraries such as "pandas" (Reback et al., 2020) and "numpy" (Harris et al., 2020). We appended the model name as attribute to the statistics to allocate each result to its modeling system. In addition, we implemented a module, which helped us to push the aggregated result to our initial database to reuse in follow-up research projects. In a similar approach to the input component, we established the connection to our designed PostgreSQL database via the package "SQLAlchemy" (Bayer, 2012).

Similarly, we used the 10000 age estimates per centimeter for calculating the sedimentation rates. Our calculation used three different approaches to calculate sedimentation rates: "naïve", "moving average over three depths", and "moving average over five depths". Table 3 lists the appropriate equations for each approach. The user can decide which one of the three approaches best applies to the individual sediment record. We summarized the output into the basic summarizing statistics (mean, median, one-sigma ranges, and two sigma ranges) accessible to the users, but added the model name and employed approach as additional attributes. If users use more than one sediment core for sedimentation rate calculation, then LANDO will automatically execute the sedimentation rate calculation in parallel using the "Dask" back-end (Dask Development Team, 2016) and the "joblib" Python package (Joblib Development Team, 2020).

**Table 3** – *Approaches to calculate sedimentation rates within LANDO. The value represents the layer of interest within a sediment core for which the calculation is necessary. Both $x_{i+1}$ and $x_{i+2}$ are the following layers, while $x_{i-1}$ and $x_{i-2}$ are the previous layers. The unit for the resulting sedimentation rate is centimeter per year [cm/yr].*

| Approach | Equation |
|---|---|
| *Naïve (default)* | $\text{sedimentation rate }(x_i) = \dfrac{\text{depth}(x_i) - \text{depth }(x_{i-1})}{\text{age}(x_i) - \text{age}(x_{i-1})}$ |

| Moving average over three depths | $\text{sedimentation rate } (x_i) = \dfrac{\text{depth}(x_{i+1}) - \text{depth}(x_{i-1})}{\text{age}(x_{i+1}) - \text{age}(x_{i-1})}$ |
|---|---|
| Moving average over five depths | $\text{sedimentation rate } (x_i) = \dfrac{\text{depth}(x_{i+2}) - \text{depth}(x_{i-2})}{\text{age}(x_{i+2}) - \text{age}(x_{i-2})}$ |

## 2.5 Evaluation of model performance

To evaluate the performance of each modeling system, we looked at three different case studies:

Case Study no. 1 - Comparison of multiple modeling systems for one sediment core with a continuously deposited sequence of dating points (*"Continuously deposited sequence"* – CS1)

Case Study no. 2 - Comparison of multiple modeling systems for one sediment core with a disturbed sequence (including inversions) of dating points *("Inconsistent sequence"* – CS2)

Case Study no. 3 - Comparison of sedimentation rate changes for multiple sediment cores (*"Multiple cores"* – CS3)

We examined both sedimentation rate and age-depth modeling results in each of the three case studies. For the first case study, we selected the sediment core EN18218 (Vyse et al., 2021) to showcase the generated output of LANDO. The 6.53 m long sediment record obtained from Lake Rauchuvagytgyn, Chukotka (67.78938° N,
168.73352° E, core location water depth: 29.5 m) during an expedition in 2018 consisted of 23 bulk sediment samples used for radiocarbon sampling. The authors determined an existing age offset of 785 ± 31 yr BP (years Before Present, i.e., before 1950 CE), which we used in our modeling process as well.

As counterexample for the second case study, we have chosen the sediment core EN18208 (Vyse et al., 2020). During the same expedition to Russia's Far East in 2018, scientists recovered this EN18208 core from Lake Ilirney,
Chukotka (67.34030° N, 168.29567° E, core length: 10.76 m, core location water depth: 19.0 m). The authors based their age-depth model on four OSL dates and 17 radiocarbon dates from bulk sediment samples as well as an age offset of 1721 ± 28 yr BP. However, in addition to the age offset, we included all seven available OSL and 25 radiocarbon dates for this core in our study.

Both cores are also part of the "*Multiple cores"* case study with a total of 55 sediment cores (Figure 1). More
details on each sediment cores are accessible in the corresponding references, which we list in Table 4.

**Table 4** – *List of all datasets used in this study. Main data source or repository are either the Pangaea database, PaleoLake database, or tables within the main body or supplementary material of publications. Data accessible links to the main data source. Paper reference includes citation to the latest version of the corresponding dataset.*

| CoreID | PaleoLake Database ID | Age-Depth Model Available | Main Data Source / Repository | Data Accessible | Paper Reference |
|---|---|---|---|---|---|

| | | | | | |
|---|---|---|---|---|---|
| 16-KP-04-L19 | | Yes | Publication | https://doi.org/10.1111/bor.12521 | Andreev et al., 2021 |
| 2008-3 | | Yes | Publication | https://doi.org/10.1016/j.quascirev.2012.06.002 | Rudaya et al., 2012 |
| BC2008 | | No | Publication | https://doi.org/10.1016/j.rg.2016.07.005 | Zhdanova et al., 2017 |
| BL02-2007 | | No | Publication | https://doi.org/10.1016/j.rg.2015.05.012 | Khazin et al., 2016 |
| BN2016-1 | | Yes | Publication | https://doi.org/10.1177/09596836211019093 | Rudaya et al., 2021 |
| Chupa-8 | 295 | No | PaleoLake DB | https://clck.ru/N5ksZ -- PALEOLAKE DATABASE ID 295 | Kolka et al., 2015 |
| Co1309 | 76 | Yes | Publication | https://doi.org/10.1111/bor.12379 | Gromig et al., 2019 |
| Co1412 | | Yes | Publication | https://doi.org/10.1111/bor.12476 | Baumer et al., 2021 |
| CON01-603-5 | | Yes | Pangaea | https://doi.pangaea.de/10.1594/PANGAEA.856103 | Piotrowska et al., 2004 |
| Dolgoe2012 | 335 | No | Publication | https://doi.org/10.7868/S0435428118020049 | Kolka et al., 2018 |
| EN18208 | | Yes | Pangaea | https://doi.pangaea.de/10.1594/PANGAEA.921228 | Vyse et al., 2020 |
| EN18218 | | Yes | Publication | https://doi.org/10.5194/bg-18-4791-2021 | Vyse et al., 2021 |
| ESM-1 | | Yes | Publication | https://doi.org/10.1016/j.quascirev.2012.03.004 | Mackay et al., 2012 |
| KAS-1 | | No | Publication | https://doi.org/10.1017/qua.2017.21 | Lozhkin et al., 2017 |
| Korzhino2010 | 336 | No | PaleoLake DB | https://clck.ru/N5ksZ -- PALEOLAKE DATABASE ID 336 | Syrykh et al., 2021 |
| LENDERY180-4 | 342 | No | PaleoLake DB | https://clck.ru/N5ksZ -- PALEOLAKE DATABASE ID 342 | Shelekhova et al., 2021b |
| LENDERY192 | 343 | No | PaleoLake DB | https://clck.ru/N5ksZ -- PALEOLAKE DATABASE ID 343 | Shelekhova et al., 2021b |
| LENDERY200-1 | 344 | No | PaleoLake DB | https://clck.ru/N5ksZ -- PALEOLAKE DATABASE ID 344 | Shelekhova et al., 2021b |

| | | | | | |
|---|---|---|---|---|---|
| LENDERY203-3 | 345 | No | PaleoLake DB | https://clck.ru/N5ksZ -- PALEOLAKE DATABASE ID 345 | Shelekhova et al., 2021b |
| LOT83-7 | 321 | No | PaleoLake DB | https://clck.ru/N5ksZ -- PALEOLAKE DATABASE ID 321 | Syrykh et al., 2021 |
| LS-9 | | Yes | Publication | https://doi.org/10.1016/S0 277-3791(00)00120-7 | Pisaric et al., 2001 |
| Maloye-1 | | No | Publication | https://doi.org/10.1017/qua .2017.21 | Lozhkin et al., 2017 |
| MC2006 | | No | Publication | https://doi.org/10.1016/j.rg g.2015.05.012 | Khazin et al., 2016 |
| Muan2018 | 339 | No | PaleoLake DB | https://clck.ru/N5ksZ -- PALEOLAKE DATABASE ID 339 | Shelekhova and Lavrova, 2020 |
| Okun2018 | 338 | No | Publication | https://doi.org/10.17076/li m1319 | Shelekhova et al., 2021a |
| OSIN | 110 | No | Publication | https://doi.org/10.17076/li m305 | Tolstobrova et al., 2016 |
| PER3 | | Yes | Publication | https://doi.org/10.1007/s10 933-015-9858-y | Anderson et al., 2015 |
| PG1111 | | Yes | Publication | https://doi.org/10.1016/j.q uaint.2004.01.032 | Andreev et al., 2004 |
| PG1205 | | Yes | Pangaea | https://doi.pangaea.de/10.1 594/PANGAEA.734962 | Wagner et al., 2000 |
| PG1214 | | Yes | Pangaea | https://doi.pangaea.de/10.1 594/PANGAEA.734137 | Cremer et al., 2001 |
| PG1228 | | Yes | Pangaea | https://doi.pangaea.de/10.1 594/PANGAEA.726591 | Andreev et al., 2003 |
| PG1238 | | Yes | Publication | https://doi.org/10.1016/S0 277-3791(03)00139-2 | Raab et al., 2003 |
| PG1341 | | Yes | Publication | https://doi.org/10.1101/202 1.11.05.465756 | von Hippel et al., 2021 |
| PG1351 | | Yes | Publication | https://doi.org/10.1046/j.1 365-246X.2002.01625.x | Nowaczyk et al., 2002 |
| PG1437 | | Yes | Pangaea | https://doi.pangaea.de/10.1 594/PANGAEA.728450 | Andreev et al., 2005 |
| PG1746 | | Yes | Pangaea | https://doi.pangaea.de/10.1 594/PANGAEA.802677 | Nazarova et al., 2013 |
| PG1755 | | Yes | Publication | https://doi.org/10.1016/j.q uascirev.2010.04.024 | Müller et al., 2010 |

| | | | | | |
|---|---|---|---|---|---|
| PG1756 | | Yes | Pangaea | https://doi.pangaea.de/10.1594/PANGAEA.708169 | Müller et al., 2009 |
| PG1856 | | Yes | Publication | https://doi.org/10.1016/j.gloplacha.2015.07.011 | Hoff et al., 2015 |
| PG1857 | | Yes | Publication | https://doi.org/10.1016/j.gloplacha.2015.07.011 | Hoff et al., 2015 |
| PG1858 | | Yes | Publication | https://doi.org/10.1007/s10933-012-9580-y | Hoff et al., 2012 |
| PG1890 | | Yes | Publication | https://doi.org/10.1016/j.gloplacha.2015.07.010 | Dirksen et al., 2015 |
| PG1972 | | No | Pangaea | https://doi.pangaea.de/10.1594/PANGAEA.780526 | Biskaborn et al., 2013a |
| PG1975 | | No | Pangaea | https://doi.pangaea.de/10.1594/PANGAEA.780385 | Biskaborn et al., 2013b |
| PG1984 | | Yes | Pangaea | https://doi.pangaea.de/10.1594/PANGAEA.776407 | Biskaborn et al., 2012 |
| PG2023 | | Yes | Pangaea | https://doi.pangaea.de/10.1594/PANGAEA.848897 | Biskaborn et al., 2016 |
| PG2133 | | Yes | Publication | https://doi.org/10.3389/fevo.2021.625096 | Courtin et al., 2021 |
| PG2201 | | Yes | Publication | https://doi.org/10.3389/feart.2021.710257 | Hughes-Allen et al., 2021 |
| PG2208 | | Yes | Publication | https://doi.org/10.3389/feart.2021.737353 | Biskaborn et al., 2021 |
| Tel2006 | | Yes | Pangaea | https://doi.pangaea.de/10.1594/PANGAEA.914417 | Rudaya et al., 2016 |
| Teriberka17 | 341 | No | Publication | https://doi.org/10.17076/lim865 | Tolstobrov et al., 2018 |
| TKT-3 | | Yes | Publication | https://doi.org/10.1016/j.quaint.2020.05.023 | Lozhkin et al., 2020 |
| TL-1-1 | | No | Publication | https://doi.org/10.1191/095968399669823431 | Wolfe et al., 1999 |
| TULOMA27 | 23 | No | Publication | https://doi.org/10.1016/S0921-8181(01)00118-7 | Corner et al., 2001 |
| UKhau2015 | 337 | No | Publication | https://doi.org/10.31857/S0869607121060070 | Shelekhova et al., 2021c |


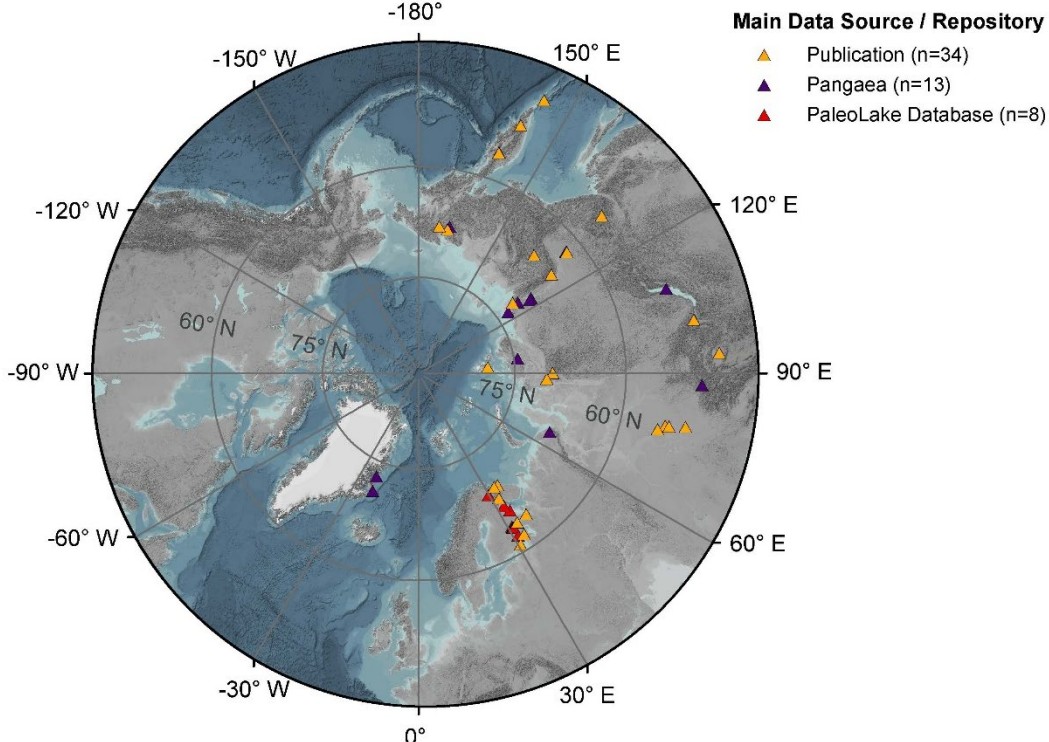

**Figure 1** – *Map of geographical distribution of lake sediment cores used for our study (triangles, n = 55). Orange triangles (n = 34) represent sediment cores for which we obtained age determination data from a related publication. Purple triangles (n=13) show datasets we collected from the publicly accessible Pangaea database (Diepenbroek et al., 2002). Red triangles (n = 8) indicate referenced datasets provided by the PaleoLake Database (Syrykh et al., 2021). ArcGIS Basemap: GEBCO Grid 2014 modified by AWI. The outer ring in the graphic corresponds to 45° N.*

### 2.5.1 Numerical combination of model outputs

To introduce the ensemble model in LANDO, we combined the outputs from all five modeling systems into one composite model. We considered the outermost limits (min. and max. values) of all confidence intervals (one-sigma or two-sigma) as our boundary for the ensemble model. By taking these outermost limits into account, we artificially increased the area of uncertainty covered by the ensemble model, but we made sure that we were representing all possible outcomes and maximizing the likelihood of including the true chronology. We also included a weighted average ($\bar{x}$) of the age estimates and sedimentation rates, which we calculated using the following equations:

$$\bar{x} = \sum_{k=1}^{m} \frac{n_k}{n} * \bar{x}_k \qquad (Eq.\,1)$$

$$n = \sum_{k=1}^{m} n_k \qquad (Eq.\,2)$$

with m being the number of participating modeling systems, n as the total number of iterations as well as $\overline{x_k}$ and $n_k$ representing the median value (either for age estimate or sedimentation rate) and the associated number of

iterations from each modeling system, respectively. In some cases, the weights from each modeling system are equal, as they produce the same number of iterations. Then we can simplify Eq. 1 to represent the arithmetic mean:

$$\bar{x} = \frac{1}{m} \sum_{k=1}^{m} \bar{x}_k \qquad (Eq.\,3)$$

For our "*Multiple cores*" case study (CS3), we additionally had to ensure comparability of sedimentation rates between sediment cores, since each model assigns a different age value to its sedimentation rate value per

centimeter. Therefore, we binned sedimentation rate results into 1000-year bins for each age-depth model as well as the ensemble model and calculated the weighted averages and their confidence intervals within these bins. Inside LANDO, users can change the initial bin size of 1000 years to the desired resolution.

### 2.5.2    Detection and filtering of unreasonable models

For cases in which age-depth models do not agree with each other, e.g., *"Inconsistent sequence"* case study (CS2),

we have built in the option of importing data from measured sediment properties, also known as proxies. Because of compositional and density variations of deposits, changes in sedimentation rates imply changes in the deposition of proxies (Baud et al., 2021; Biskaborn et al., 2021; Vyse et al., 2021). By including appropriate, independent proxy data on lithological changes within the sediment core, we can weight each model based on its performance to represent these variations in sedimentation rate. Users should provide the independent sediment proxy data as

file with two columns, namely "compositedepth" which should be the measurement depths (as mid-point centimeter below sediment surface), and "value" representing the values of the proxy. This simplification makes it possible to import different available proxies or statistical representations of proxy data, i.e., results from ordination techniques (PCA, MDS, etc.), into the optimization process and to visualize the behavior of the age-depth models in comparison to these proxies.

In order to evaluate the performance, we adapted the fuzzy change point approach by Hollaway et al. (2021) to work with our input data and desired outcome on a depth-dependent scale instead of a time series. Similarly to Hollaway et al. (2021), our approach firstly detected change points within the proxy data and each modeling system output by fitting an ARIMA model to the data and then extract change points by using the "changepoint" R package (Killick and Eckley, 2014; Killick et al., 2016) on the residuals of the ARIMA model. If we found no change points

in the proxy data via this approach, we applied the "changepoint" R package on the raw independent sediment proxy data instead. Through the additional bootstrapping process introduced by Hollaway et al. (2021), we were able to set up confidence intervals for the extracted change points. Subsequently, we searched for the intersection between the change points plus their confidence interval for each age-depth model with the independent proxy data. After converting the change points for both age-depth model and independent proxy data into triangular fuzzy

numbers, we obtained similarity scores using the Jaccard similarity score of the fuzzy number pairs as described in Hollaway et al. (2021). The similarity score can reach numbers between zero (no match) and one (perfect match). However, the threshold of excluding an age-depth model from the generated combined model depends on the imported proxy data and number of detected change points. Therefore, the user can set the threshold accordingly to their proxy within LANDO, but we have implemented the default value for this threshold to 0.1, which

corresponds to an overlap of 10% of the change points between model and proxy data.

In addition to the criterion of preparing the proxy data in the format of "depth vs. value" in a separate file, we suggest using a proxy with a high resolution. As a high-resolution proxy, we define a proxy with more than 50 measurements per meter of core length. For our *"Inconsistent sequence"* case study (CS2), we used high-resolution elemental proxy data from XRF (X-ray fluorescence) measurement as our independent proxy data. As our evaluation element to optimize the age-depth models, we selected zircon ("Zr"), which itself is an indicator for minerogenic/detrital input (Vyse et al., 2020 and references therein). The zircon proxy data of EN18208 has a resolution of 200 measurements per meter of core length.

To achieve a realistic comparison between sediment cores in the *"Multiple cores"* case study (CS3), we looked at the individual age-depth model outputs for each sediment core to determine whether an optimization step was required. We have only selected sediment cores with a published age-depth model (n = 33) so that we can refer to lithological boundaries from the original publication. During the analysis, we saw that nine sediment cores needed to be optimized due to strong inconsistencies between models over the entire length of each core. In twelve cases, where models within the lower section of the cores did not match, we considered proxy-based optimization to improve the model outcome when high-resolution data was available.

### 2.5.3 Display of models

To display the results from age-depth modeling and sedimentation rate calculation, we decided to create our own plots, instead of reusing the plots from each individual modeling system. Our plot header contains the unique CoreID; additionally, the header indicates whether the user decided to apply a reservoir correction on the radiocarbon data or not. Our single core plots consist of two main panels: On the left-hand side, the panel shows the results from the age-depth modeling process with the calibrated ages (in calibrated years Before Present, i.e., before 1950 CE) on the x-axis and the composite depth of the sediment core (in centimeter) on the inverted y-axis. On the right-hand side, the panel displays the result from the sedimentation rate calculation (in cm/yr, centimeter per year) on the x-axis plotted against the same composite depth on the inverted y-axis. For better readability of the strong variability of sedimentation rate, we used the log scale for the x-axis of the right panel. Generally, LANDO draws the ensemble age-depth model and sedimentation rate in grey with the weighted average as dashed line.

For all models, LANDO will display the median values for age and sedimentation rate as solid lines. Both panels further display the corresponding one-sigma range and two-sigma range per centimeter for each model. Depending on the user's selection, users can plot both sigma ranges, only one of the two sigma ranges, or just the median ages. To include age determination data within the plots, LANDO internally calibrates the radiocarbon data with the "BchronCalibrate" function of the *Bchron* package (Haslett and Parnell, 2008; Parnell et al., 2008) with either the IntCal20 (Reimer et al., 2020), Marine20 (Heaton et al., 2020), or SHCal20 (Hogg et al., 2020) calibration curve. This allows users to analyze samples from locations other than the terrestrial northern hemisphere. By default, the left panel contains each age data point as a predefined symbol with its one-sigma uncertainty as error bar. The symbol used by LANDO depends on the material category defined in the input file for each dating point.

If users decide to filter out unreasonable age-depth models, similar to *"Inconsistent sequence"* case study (CS2), we added the option to plot the independent proxy data and therefrom derived lithology as an additional panel on the left-hand side for a better interpretability. Further, LANDO highlights the boundaries of lithological change

and its confidence interval in both sedimentation rate and age-depth model plots. The optimized plot includes a goodness-of-fit for each involved modeling system to represent the change points at the bottom of the plot.

When using LANDO for multiple sediment cores, the overall plot holds for each sediment core the results from the binned weighted average sedimentation rate calculation (as median sedimentation rate in cm/yr, centimeter per year) against the selected age bins (in calibrated years Before Present, i.e., before 1950 CE) for each modeling system. This visual illustration allows user to compare multiple sediment cores based on the time axis.

For people with color vision deficiency, we incorporated the extra option to plot the resulting age-depth plots with different line styles and textures to support the visual differentiation between each model. Figure S4 in the supplementary material shows the color-blind friendly output created by LANDO. With LANDO we want to support inclusivity in science, but we look forward to feedback from the community on how we can improve LANDO in this regard.

### 2.6 Further analysis – Sedimentation rate development over time

To identify similar temporal shifts in sedimentation regimes in our case study *"Multiple cores"* (CS3), we examined our data collection of 55 sediment cores regarding a general tendency in sedimentation rate shifts. First, we considered the 11 700-yr BP (Before Present, i.e., before 1950 CE) boundary as our marker for the change between Holocene and Late Pleistocene to separate the datasets (Rasmussen et al., 2006; Lowe and Walker, 2014; Walker et al., 2008). We selected this marker because numerous studies suggest a general difference in sedimentation regimes between these periods (e.g., Baumer et al., 2021; Bjune et al., 2021; Kublitskiy et al., 2020; Müller et al., 2009; Wolfe, 1996; Vyse et al., 2021). As some of the models were below the 11 700-yr BP marker, the calculation of the mean sedimentation rate for the Late Pleistocene featured only a subset of sediment cores (total number of sediment cores with measurement in Late Pleistocene: 20). Then, for each age model of the sediment cores in the subset, we used the two-sigma ranges around 11 700 yr BP to determine whether the maximum absolute change occurred exactly at 11 700 yr BP or around our set marker. For this investigation, we changed the bin size to 100-year bins to allow comparison between each modeling system and the combined models. Using maximum from the interquartile ranges of the two-sigma ranges for each model (see supplementary material Figure S3), we defined the observation period from 8700 to 14 700 yr BP (corresponds to a range of ± 3000 years). We then checked the data within the time span to see where the maximum change in sedimentation rate occurred. If the calculated age for the new marker was at the edge of our time span, we iteratively increased the outer limit by 100 years (up to a maximum of 18 000 yr BP) to see if the calculated age still reflected the maximum absolute change. We then used the newly defined marker to calculate the mean sedimentation rate for before and after the marker.

### 3 Results

### 3.1 "Continuously deposited sequence" – Case Study no. 1

All five age-depth models were able to produce an age-depth relationship for sediment core EN18218 ("Lake Rauchuvagytgyn") with only small diversions in between some of the calibrated ages. Figure 2 depicts the two

visual outputs produced by LANDO. Panel (a) displays all models side by side, while panel (b) shows the combined output from all models.

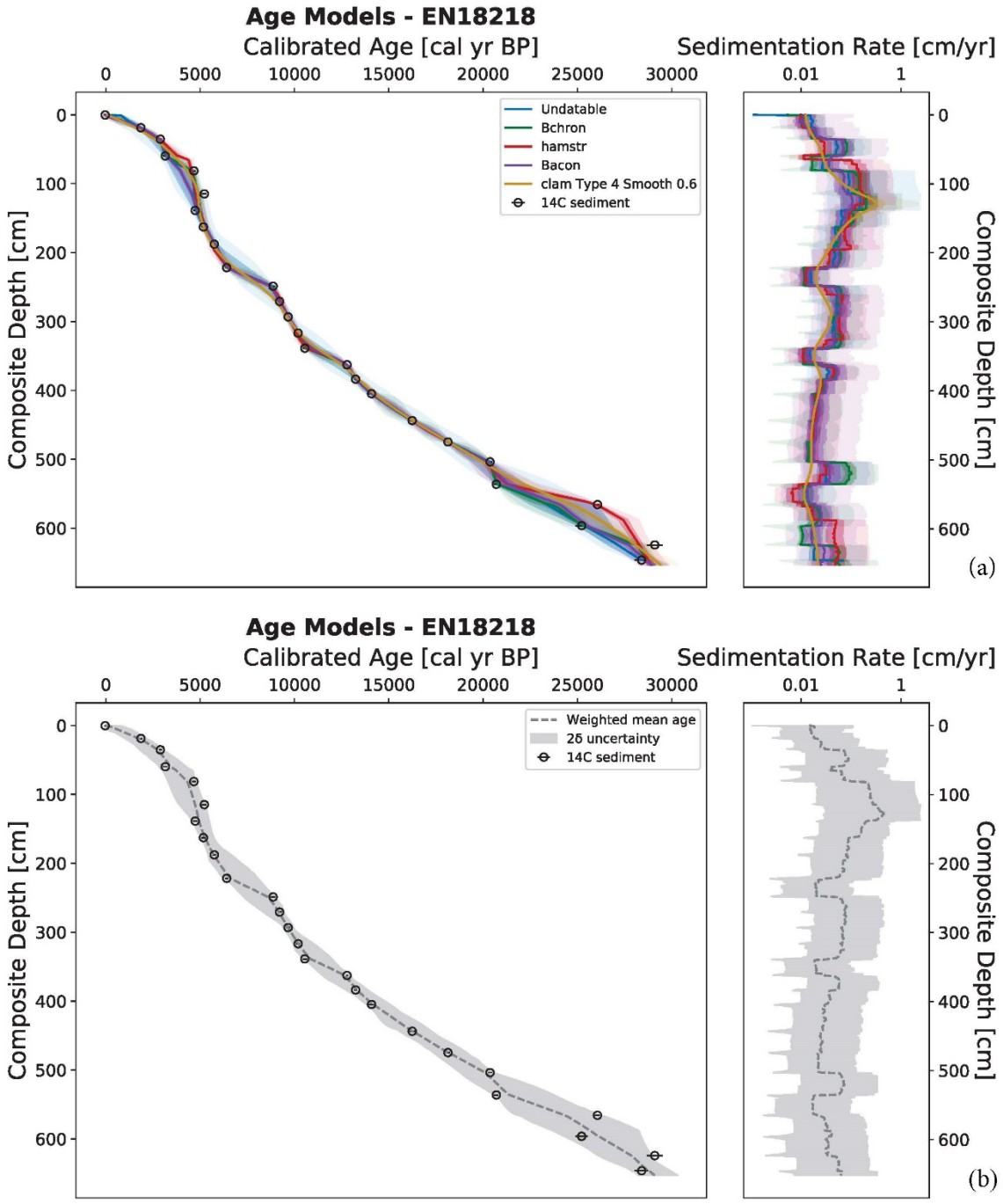

**Figure 2** – *Generated output from LANDO for sediment core EN18218 ($^{14}C$ data from Vyse et al., 2021) as an example of continuous lacustrine sedimentation over time. Panel (a) consists of a comparison between age-depth*

*models from all five implemented modeling systems (left plot) and their calculated sedimentation rate (right plot). Colored solid lines indicate both the median age and median sedimentation rate for all models, while shaded areas represent their respective one-sigma and two-sigma ranges in the same colors with decreasing opacities. Panel (b) shows the ensemble age-depth model (left plot) and its sedimentation rate (right plot). The dashed line in panel (b) represents the weighted average age estimates (left plot) and the weighted average*

 *sedimentation rates (right plot) for the ensemble model, while the grey area represents the two-sigma*
*uncertainty, i.e., the outermost limits of two-sigma ranges from all models. Both plots on the left of (a) and (b)*
*show the depth below sediment surface on the inverted y-axis as composite depth of the sediment core in*
*centimeter (cm) and the calibrated ages on the x-axis in calibrated years Before Present (cal. yr BP, i.e., before*
*1950 CE). Black circles within (a) and (b) indicate the calibrated $^{14}C$ bulk sediment samples with their mean*

*calibrated age using the IntCal20 calibration curve (Reimer et al., 2020) and their one-sigma uncertainty as*
*error bars. The plots on the right display the sedimentation rate in centimeter per year (cm/yr, x-axis as log-*
*scale) against the depth below sediment surface as the composite depth of the sediment core in centimeter (cm,*
*inverted y-axis).*

All models revealed highest sedimentation rates for the interval between 108 cm and 133 cm. Mean values ranged

from 0.242 cm/yr (*hamstr*) to 0.764 cm/yr (*clam*) within this interval, whereas the median sedimentation rate varied between 0.107 cm/yr (*Bacon*) and 0.314 cm/yr (*clam*). In the lower segment of EN18218 (653 cm to 504 cm), the models showed a stronger disagreement among each other with larger varying mean and median values for sedimentation rate. In three instances, the majority of models noticeable dropped to lower sedimentation rate values. We found the first two declines in sedimentation rate between 366 cm and 339 cm as well as between 249

cm and 222 cm with median sedimentation rates from 0.012 cm/yr (*hamstr*) to 0.027 cm/yr (*Bacon*) and from 0.013 cm/yr (*hamstr*) to 0.025 cm/yr (*Bacon*), respectively. The last significant downward shift occurred between 66 cm and 57 cm, where *hamstr* decreased the median sedimentation rate tenfold from 0.15 to 0.015 cm/yr between 66 cm and 64 cm.

In our ensemble model, we found the highest value for weighted average sedimentation rate at 128 cm with 0.4483

cm/yr (two-sigma range: 0.032 - 2.338 cm/yr), which corresponded to weighted average age estimate of 4846 cal yr BP (two-sigma range: 4301 - 5384 cal yr BP). Throughout the core, the cumulative two-sigma uncertainty of the ensemble model ranged from 0.002 cm/yr to 2.486 cm/yr.

**3.2 "Inconsistent sequence" – Case Study no. 2**

For the second case study, four out of five modeling systems produced an output for sediment core EN18208

("Lake Ilirney"). The modeling system *clam* was unable to produce an age-depth model for this core. Figure 3 shows the visual outputs with all models in panel (a) and the combined model in panel (b). Figure 4 consists of three panels showing the results from the proxy-based optimization process using zircon (Zr). Panel (a) shows the visual output from the optimization process, while panel (b) and (c) illustrate the optimized age-depth model with the highest matching score and the resulting ensemble model, respectively.

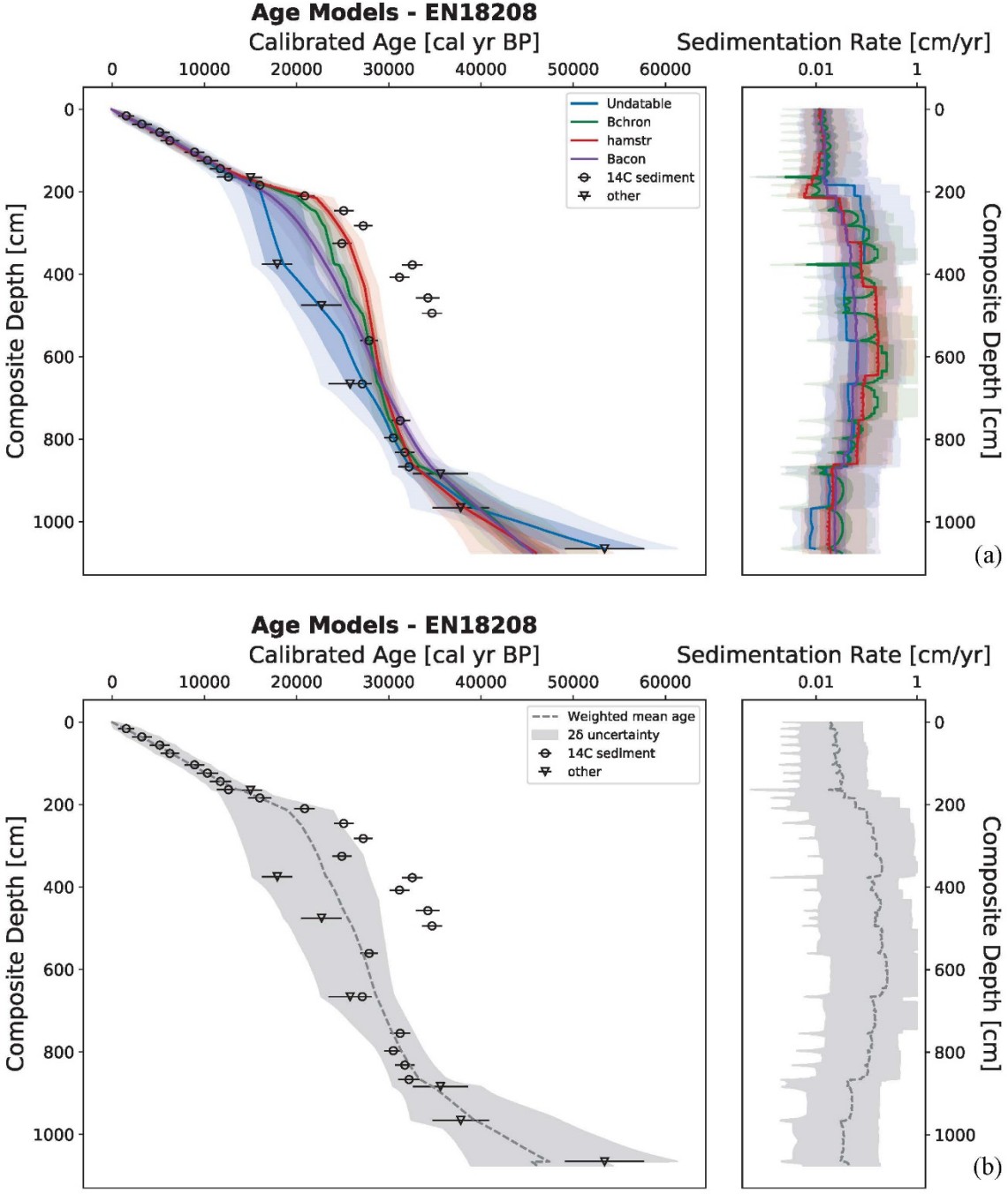

**Figure 3** – *Generated output from LANDO for sediment core EN18218 (OSL and [14]C data from Vyse et al., 2020) as an example of discontinuous lacustrine sedimentation. Panel (a) consists of a comparison between age-depth models from four out of five implemented modeling systems (left plot) and their calculated sedimentation rate (right plot). The modeling system clam was unable to produce an age-depth model for this core. Colored solid lines indicate both the median age and median sedimentation rate for all four models, while shaded areas represent their respective one-sigma and two-sigma ranges in the same colors with decreasing opacities. Panel (b) shows the ensemble age-depth model (left plot) and its sedimentation rate (right plot). The dashed line in panel (b) represents the weighted average age estimates (left plot) and the weighted average sedimentation rates (right plot) for the ensemble model, while the grey area represents the two-sigma uncertainty, i.e., the outermost limit of two-sigma ranges from all four models. Both plots on the left of (a) and (b) show the depth below*




*sediment surface on the inverted y-axis as composite depth of the sediment core in centimeter (cm) and the calibrated ages on the x-axis in calibrated years Before Present (cal. yr BP, i.e., before 1950 CE). Black circles within (a) and (b) indicate the calibrated $^{14}C$ bulk sediment samples with their mean calibrated age using the IntCal20 calibration curve (Reimer et al., 2020) and their one-sigma uncertainty as error bars. Black down-pointing triangles show mean ages from OSL analysis and their one-sigma uncertainty as error bars. The plots on the right display the sedimentation rate in centimeter per year (cm/yr, x-axis as log-scale) against the depth below sediment surface as the composite depth of the sediment core in centimeter (cm, inverted y-axis).*

While *Undatable* was the only modeling system that considered the dating point at 1066 cm before following the next dating point at 966 cm, all remaining three modeling systems assumed a steady accumulation (mean sedimentation rate: 0.0575 cm/yr) from 1076 cm before overlapping their paths with *Undatable*. At the depth of 795 cm, we found the next divergence between the age-depth models. *Undatable* followed the younger OSL dates and the young radiocarbon date at 666 cm. *Bacon*, *Bchron*, and *hamstr* continued with the radiocarbon date at 561 cm, before taking different paths until age determination point at 184 cm. All modeling systems again overlapped their paths from 184 cm to the sediment surface with a mean sedimentation rate of 0.0277 cm/yr.

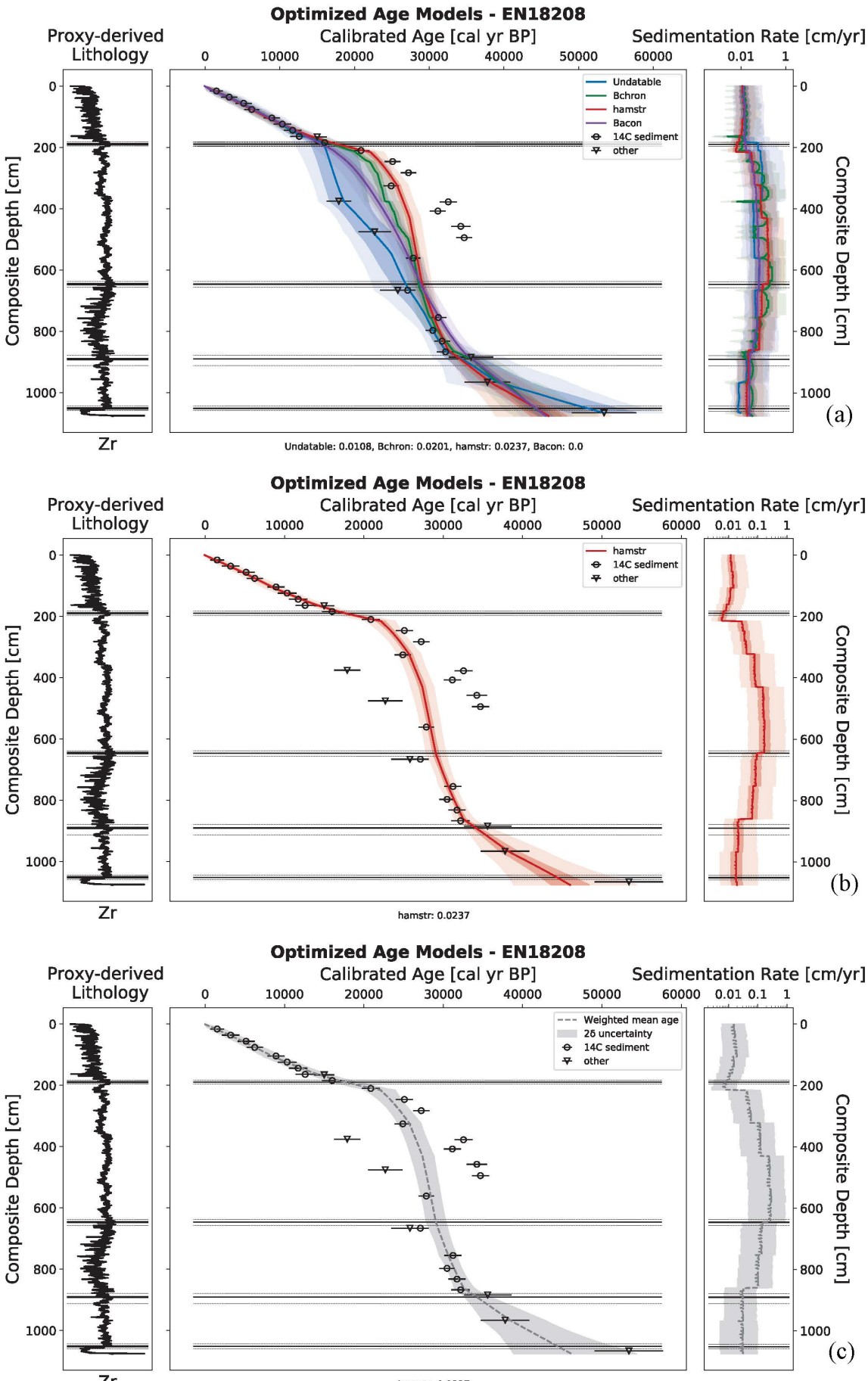

**Figure 4** – *Optimized visual output for EN18208 (OSL and $^{14}C$ data from Vyse et al., 2020). We used high-resolution X-ray fluorescence (XRF) measurements of zircon (Zr) as independent proxy to evaluate model performance to represent lithological changes. Panel (a) extends the existing panel (a) of Figure 3 by adding a plot on the left to show the proxy-derived lithology used to filter unreasonable models. This added plot consists of the proxy measurements of Zr (in counts per second) along the depth below sediment surface as the composite depth of the sediment core in centimeter (cm) and the derived lithological boundaries (solid horizontal lines) plus their uncertainty range (dashed horizontal lines). Both age-depth model and sedimentation rate plot contain the same lithological boundaries as visual aid. The text box in the bottom middle lists the models with their matching score related to the proxy-derived lithology. Panel (b) shows the model (hamstr) with the highest matching score (0.0237). Panel (c) depicts our ensemble model based on this model. The age-depth models displayed in panel (b) and (c) show strong similarities with the age-depth model developed by Vyse et al. (2020).*

During the optimization process, our adapted algorithm located four lithological boundaries with its uncertainty range from the independent proxy data: 189.5 cm (182 – 192.5 cm), 646 cm (638 – 657 cm), 890.5 cm (874 – 912 cm), and 1051.5 cm (1043 – 1061.5 cm). We found the highest matching score from the optimization for *hamstr* (Score: 0.0237). Table 5 shows the average sedimentation rate for each proxy-derived lithological unit (PLU) of the ensemble model of EN18208.

**Table 5** – *Average sedimentation rate of EN18208 divided into proxy-derived lithological units. The calibrated mean model range indicates the mean age estimates of the ensemble model for the corresponding depths of the proxy-derived lithological unit (PLU).*

| Proxy-derived lithological unit | Corresponding depths below sediment surface [cm] | Calibrated mean model range [cal yr BP] | Average sedimentation rate [cm/yr] |
|---|---|---|---|
| PLU1 | 0 – 190 | -67 – 17752 | 0.0152 |
| PLU2 | 190 – 646 | 17752 – 29073 | 0.1664 |
| PLU3 | 646 – 891 | 29073 – 34244 | 0.1073 |
| PLU4 | 891 – 1052 | 34244 – 44499 | 0.0307 |

**3.3 "Multiple cores" – Case Study no. 3**

In contrast to the previous case studies, this case study focused on understanding the development of sedimentation rates over time, with the emphasis on the transition from the Holocene to the Pleistocene. We used age determination data from 33 sediment cores with a published age-depth model to show the standard output of LANDO for multiple sediment cores, while using all datasets for the subsequent analyses. Figure 5 shows the ensemble models with weighted average sedimentation rates binned into 1000-year bins from our multi-core investigation with 33 published sediment cores (see Figure S1 for the individual models in the supplementary material). We set the boundaries from 0 to 21 000 cal yr BP within these figures to cover the time span from the present to the Last Glacial Maximum (LGM) (Clark et al., 2009). Below the number for each core in Figure 5 are the proxies used for their optimization. In 17 out of 55 cases within our entire collection, the ensemble model was based on four out of five models, as neither *clam* or *Undatable* was able to find a suitable age-depth model (for more details, please see Table S1 in the supplementary material). The maximum time span covered by the sediment cores varied between 2000 yr BP (CoreID: PG1972) and 320 000 yr BP (CoreID: PG1351). The average non-

optimized sedimentation rate ranged between 0.004 cm/yr (CoreID: LOT83-7) and 1.142 cm/yr (CoreID:
PG1228). In total, we optimized seven sediment cores, as in most cases neither high-resolution data was available
nor the provided proxy data represented a lithological proxy when crosschecked with the original publication.
From these seven sediment cores, we reconstructed the proxy-based lithology twice with TOC as a low-resolution
proxy (CoreID: PG1228 & PG1437).

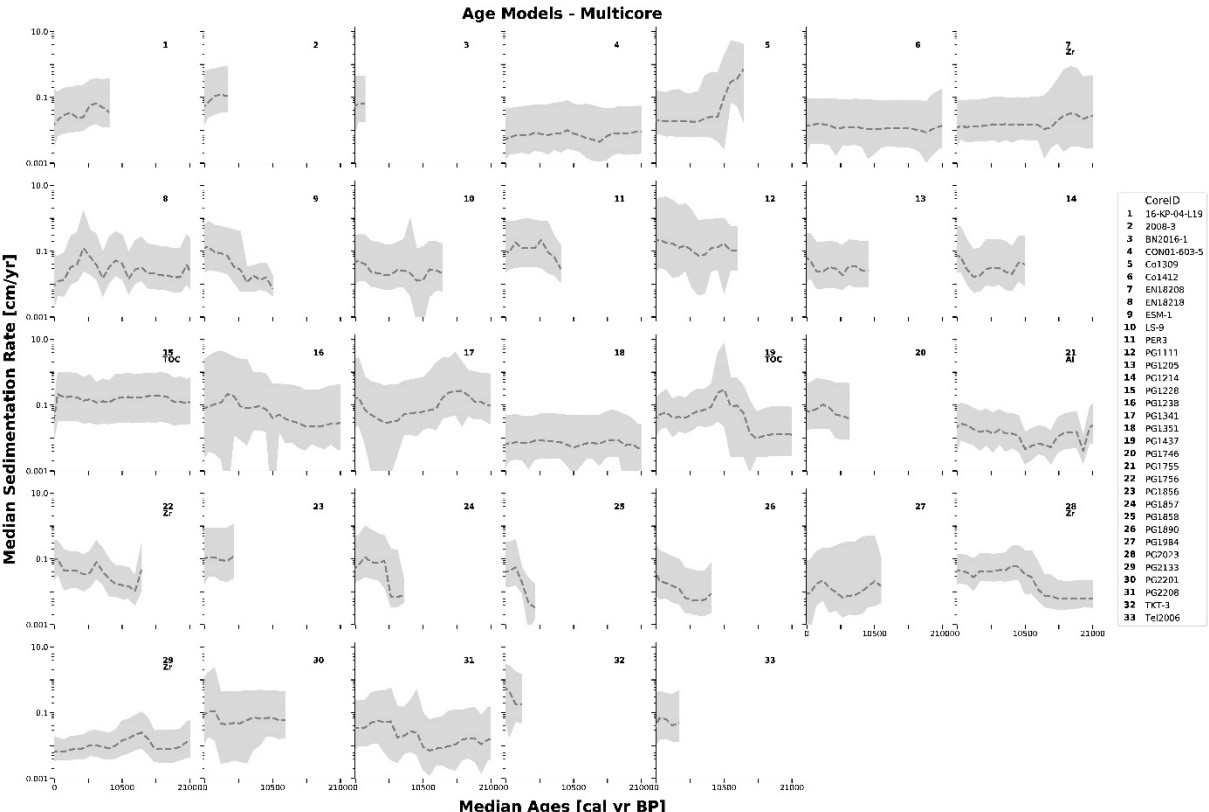

**Figure 5** – *Optimized combined models for 33 sediment cores with a published age-depth model displayed as*
*weighted average sedimentation rate (in centimeter per year, cm/yr – y-axis) binned into 1000-year bins (in*
*calibrated years Before Present, cal. yr BP, i.e. before 1950 CE – x-axis) for the last 21 000 years. Dashed line*
*represents the weighted average sedimentation rate, whereas the grey areas are the respective two-sigma*
*ranges. Each grid cell contains the unique core identifier of each involved sediment core. In seven cases, the*
*letters below each number give the name of the independent proxy used for optimization process.*

To visualize the difference in sedimentation rates between two neighboring and fundamentally different
environmental settings, i.e. Pleistocene glacial and Holocene interglacial, we used the datasets that were split at
the Holocene-Pleistocene boundary at 11 700 yr BP. Figure 6 shows the mean sedimentation rate for Holocene
and Late Pleistocene for each model with its one-sigma uncertainty. Figure S3 in the supplementary material gives
an overview over the overall uncertainty for all models. Among all models, *clam* models have the lowest range on
average for both Holocene (0.0135 cm/yr) and Late Pleistocene (0.0011 cm/yr), while the combined models show
the greatest uncertainty on average in the Holocene (0.0942 cm/yr) and for the Late Pleistocene (0.0711 cm/yr).
The sediment core PG1228 (latitude: 74.473° N) showed the highest individual sedimentation rate for the Holocene
in *Undatable* (median sedimentation rate: 1.1013 cm/yr). We observed a significant reduction of about 77 % for
the optimized model of the same core (0.1264 cm/yr), compared to its combined model (0.5615 cm/yr).

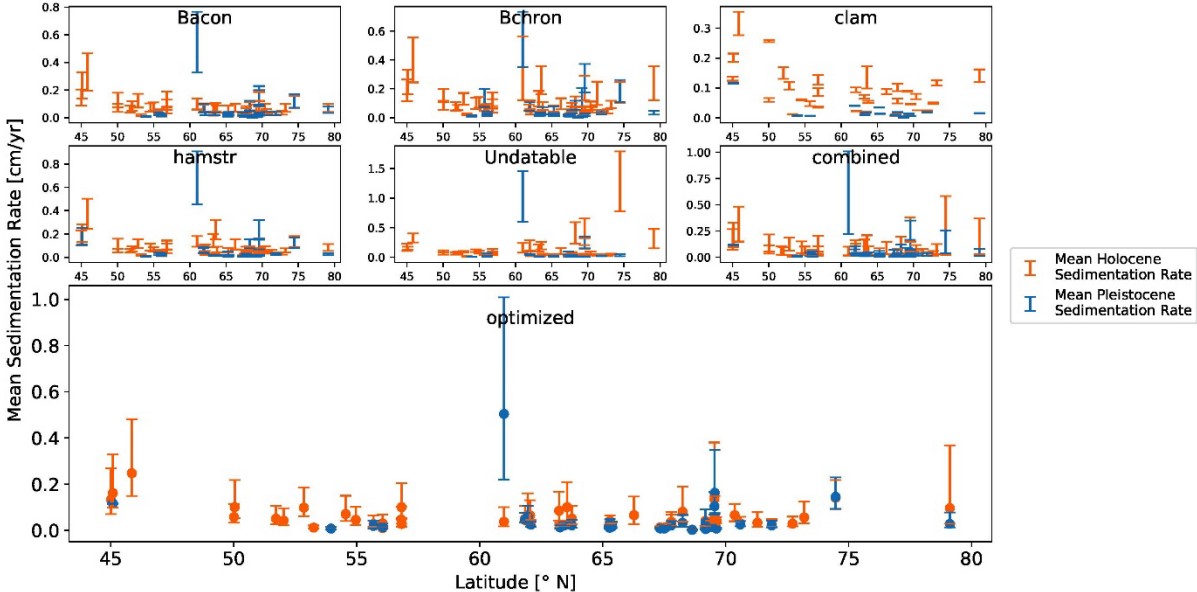

**Figure 6** – *Average sedimentation rate in centimeter per year (cm/yr) for each sediment core in our data collection of 55 sediment cores divided into Holocene dataset (from present to 11 700 yr BP, orange lines) and Late Pleistocene dataset (from 11 700 yr BP to 21 000 yr BP, blue lines). Each plot displays the one-sigma range of sedimentation rate within each dataset for each model and sediment core. In addition, filled circles represent the mean value for the optimized models.*

For our data compilation, we found the largest absolute change in sedimentation rates within the modeling systems on average between 9600 and 11 900 yr BP (Figure 7). For our combined and optimized models, however, the largest change averaged between 10 500 yr BP and 10 700 yr BP. Still, all sediment cores covered the entire range of our initial time span from 8700 to 14 700 yr BP within the models. Using the results of the largest change in sedimentation rate for each sediment core and model as new markers, we again split the datasets into two separate datasets. One dataset contained mostly Holocene sedimentation rate values (Holocene dataset), while the other contained mostly Late Pleistocene values (Late Pleistocene dataset). Therefore, the initial display (Figure 6) changed slightly to Figure 8. Most notable was the increase in total number of sediment cores in Late Pleistocene dataset with an individual separation (n = 38) compared to the Late Pleistocene dataset with the separation at 11 700 yr BP (n = 19).

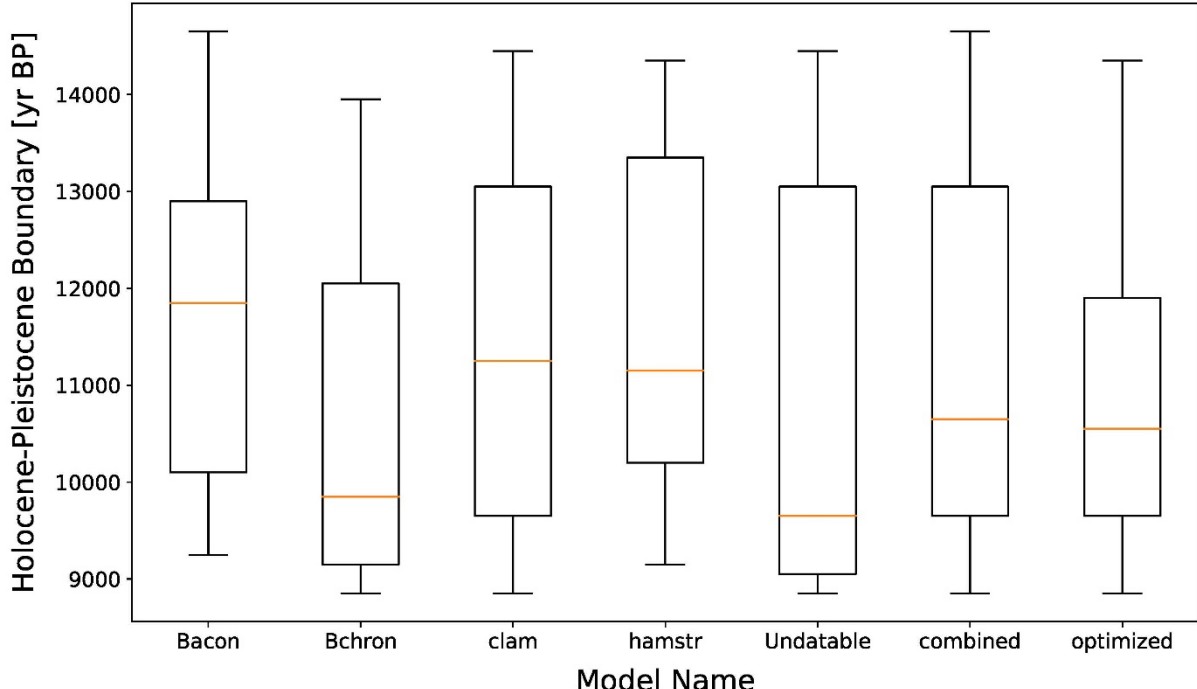

**Figure 7** – *Boxplot representing the years with the biggest absolute change in sedimentation rate for our data collection of 55 sediment cores. Sedimentation rate results from each model binned into 100-year bins to allow comparisons between the modeling systems. The initial observation time span covers 8700 to 14 700 yr BP. The orange line corresponds to the median value for each model.*

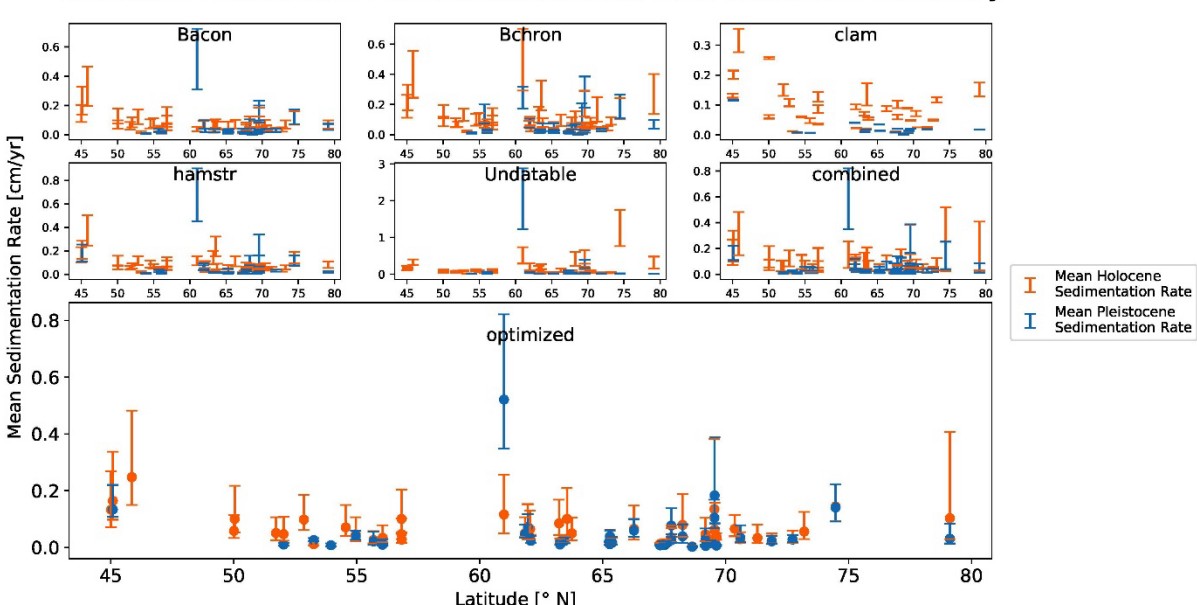

**Figure 8** – *Average sedimentation rate in centimeter per year (cm/yr) for each sediment core in our data collection of 55 sediment cores divided into Holocene dataset (orange lines) and Late Pleistocene dataset (blue lines). The exact value for the split of the datasets for each individual core and each model depends on the results of the maximum change in sedimentation rate within the observation period 8700 to 14 700 yr BP. Each*

*plot displays the one-sigma range of sedimentation rate within each dataset for each model and sediment core. In addition, filled circles represent the mean value for the optimized models.*

## 4    Discussion

### 4.1 Assessment of different case studies

By comparing the cases for the two single sediment cores, it becomes clear how age-depth relationships may diverge depending on the individual modeling system and its treatment of available dating points (cf. Wright et al., 2017; Trachsel and Telford, 2017; Lacourse and Gajewski, 2020). In the case of EN18218 (*"Continuously deposited sequence"* – CS1), all five implemented modeling systems yield an agreeing and continuous chronology. However, the two radiocarbon dates at 81.25 cm and 114.75 cm have significant impact on the model's interpretation for these depths. Vyse et al. (2021) argued that these two dates are outliers resulting from reworking and mixing effects within the sediment column. According to the authors, no additional proxy data from EN18218 would support the immediate increase in sedimentation rate for these depths and hence, they excluded both dates from the modeling process. Because we are not considering any additional proxy data to evaluate age-depth models in their geoscientific context, but rather include all provided age determination data into the modeling process, the consideration of these two radiocarbon dates on the basis of all available models leads to higher sedimentation rate. Nonetheless, the example here shows how the comprehensive application of the different modeling systems may help to identify doubtful dating points.

We saw a disagreement between the modeling systems in the case of sediment record EN18208 *("Inconsistent sequence"* – CS2), which we expected prior to the execution of our application, due to the scattered dating points in the original data. Vyse et al. (2020) linked this scatter of age data points observed in the interval between 282 and 755 cm of EN18208 to the redeposition of older carbon. They implied that to produce reliable age-depth model they had to exclude both OSL and radiocarbon dating points for these depths. However, our optimized combined model agrees with their established age-depth model and can reproduce the characteristics of the existing model by Vyse et al. (2020), without removing dating points. In addition, in three out of four cases, our proxy-derived lithology with its uncertainty matches the lithological boundaries set by the authors of the EN18208 study, according to criteria based on acoustic sub-bottom profiling. Only the first original boundary (196 cm) is outside our confidence interval from 182 cm to 192 cm. We still showed that our approach could set logical boundaries for sediment cores by solely relying on high-resolution proxy data.

Despite a strong similarity between our optimized model and the existing model developed by Vyse et al. (2020), the highest score showed a low similarity value (0.0237) using our similarity scale from zero (no match) to one (perfect match). Although we chose the highest matching score to demonstrate LANDO's ability of filtering out disagreeing models, we do not support the strategy of choosing a single age-depth model with such a low matching score. Rather, users should investigate the cause of the scatter in the age determination data and/or change the default values within LANDO. For example, to deal with the scatter in the data, users can increase the *Undatable* parameter "bootpc" to a higher value - as suggested by Lougheed and Obrochta (2019) - to account for a higher uncertainty in the given data.

Given a dataset with scatter dating points, users should also consider that models with small confidence intervals may underestimate the potential variability of their studied system. There are valid reasons for using models with

a low uncertainty, especially when the given data support such interpretations. However, although the results of these low-uncertainty models may reflect the true chronology in similar case to EN18218 *("Inconsistent sequence"* – CS2), we recommend using the combined LANDO model to include all possible outcomes. A larger uncertainty band reduces the tendency to choose a model that fits a particular hypothesis. For palaeoenvironmental reconstruction, users can also propagate these increased uncertainties into their proxy interpretation, which is often underrepresented (Lacourse and Gajewski, 2020; McKay et al., 2021).

Even though LANDO can produce age-depth models for multiple sediment cores *("Multiple cores"* – CS3), we must assume limitations in the geoscientific validity for some of the results. In a few cases, an optimization of age-depth models with independent proxy data would have been necessary, but such independent data were inaccessible or did not exist. As for these cases age-depth relationships between implemented modeling systems seem to disagree (see Figure S1 in the supplementary material), the results from our combined model might over- or underestimate the true sedimentation rate. On the other hand, optimization using proxy data can reduce these biases.

For instance, during the examination of the Holocene and the Pleistocene sedimentation rates (Figure 6), we noticed that one sediment core (PG1228) had an extremely high mean sedimentation rate for the Holocene dataset in *Undatable*. Similar to the second case study *("Inconsistent sequence"* – CS2), we found scattered age data points for this sediment core, which influenced the modeling process of *Undatable*. Further, the result then affected our combined model by increasing the overall sedimentation rate for the Holocene in this core. However, LANDO identified the *Undatable* model as an outlier based on the lithology established through independent TOC proxy data. The optimized model then agreed well with the original publication by Andreev et al. (2003), which further increased the validity of our approach. Our findings suggest that high-resolution proxy data should accompany geochronological studies to enable a more concise and realistic assessment of the development of sedimentation rates over time in high latitude lake systems.

We further improved the validity of some results of our multi-core study by comparing our LANDO output with the available age-depth models from publications. In four cases (CoreID: 2008-3, Co1309, LS-9, PG1205), we adjusted our initial output to the previously published age-depth models (Rudaya et al., 2012; Gromig et al., 2019; Pisaric et al., 2001; Wagner et al., 2000). One reason for the discrepancy was that the age determination data were not available for the entire length of sediment cores and LANDO extrapolated beyond these dating points to match the core length. In the case of PG1205 (Wagner et al., 2000) with a core length of 9.85 m, dating points were available for the upper 2.5 meters (Table 4) and therefore LANDO extrapolated the remaining seven meters to cover the entire sediment core. However, the extrapolated results in accumulation rates do not reflect the geological history of the lake record provided by Wagner et al. (2000). We have therefore changed the length of the sediment core to the last dating point to avoid strong extrapolation. In case of Co1309 (Gromig et al., 2019), the age-depth model required the introduction of a hiatus that would span from 14 to 80 cal yr. BP (Andreev et al., 2019; Savelieva et al., 2019). However, while a specific customization (such as a hiatus) is possible for single core cases, this is not possible in the current version of LANDO for multi-core investigation. To overcome this, we reduced the length of the record used in our study for core Co1309 to the depth of the last available dating point (Table 4), such that the LANDO output matches the age-depth relationship reported by Gromig et al. (2019).

The detection of sedimentation rate change as indicator for the Holocene-Pleistocene boundary yielded contrasting results. While the results from *hamstr* were closest to the 11 700-year boundary, all other modeling systems place the largest change in sedimentation rate either before or after 11 700 yr BP. We hypothesize that three factors may have influenced all model results. (1) The age uncertainty (one-sigma range) within each individual model varied on average between 1000 and 3000 years for the period of 11 600 to 11 800 yr BP (Figure S3 in the supplementary material). This wide range of uncertainty does not provide confidence in pinpointing the boundary to an exact time slice. We expect that a higher amount of dating points close to the Holocene-Pleistocene boundary could constrain the models (Blaauw et al., 2018; Lacourse and Gajewski, 2020; Trachsel and Telford, 2017), which would lead to a better estimate of the boundary. (2) The age output for each model is not evenly distributed, which means that in the period from 11 600 to 11 800 yr BP there are different numbers of observations for each core and each modeling system. We took this behavior into account by using binning (Alasadi and Bhaya, 2017). Otherwise, an interpolation between both age and sedimentation rate values could lead to potential biases in the interpretation. (3) While we assumed in our first setup that the main sedimentation rate change would occur at 11 700 yr BP consistently for all sediment cores (Figure 6), we cannot rule out the possibility that the sedimentation rate has changed significantly at different times for different lake systems. As our data collection covers a large area both in latitude and longitude (Figure 1), the variability between the models indicate the local variability between the climate and lithological preferences of the lake catchment for the involved sediment cores (e.g., Lozhkin et al., 2018; Finkenbinder et al., 2015; Anderson and Lozhkin, 2015; Kokorowski et al., 2008; Biskaborn et al., 2016; Courtin et al., 2021).

**4.2 Design of LANDO**

From the beginning of the development of LANDO, we decided to integrate most of the default settings for each modeling system as default values (Table 2). Regional studies, such as the one performed by Goring et al. (2012), have shown that specific prior information for the Bayesian modeling systems are needed to best fit the models to lakes within a geographical area. Without this regional information, changing settings within the modeling system to an arbitrary higher or lower value without considering the regional diversity could lead to under- or overfitting, if the constraints are too loose or too strict (Trachsel and Telford, 2017). For the special case that users have in-depth knowledge for one lake or multiple lake system, users can easily adapt these parameters within LANDO, as we have made these settings accessible in the Jupyter Notebook itself.

Part of the reason we made this decision was that we acquired external age determination datasets where we may not necessarily have all the essential information to specify each model. But we also wanted to simplify the process for users who do not have in-depth modeling knowledge. By using the default values, we can compare models based on their ability to work with the available data. On the other hand, we are sure that the developers have set their default values based on systematic testing. Since we did not tune the age-depth models to the existing core, i.e. changing the parameters within each modeling system, we generated "uninformed" models that solely work with the available age determination data. By combining these "uninformed" models into one model, we have created an ensemble model that we consider to be data-driven and "semi-informed".

The advantage of this data-driven, semi-informed model approach is that we are reducing the risk of overfitting by considering the uncertainty of all modeling systems. This allows us to reevaluate existing geoscientific interpretations with larger uncertainty by taking advantage of the ensemble outcome. Additionally, we found that

the more information is accessible to generate age-depth models, the more accurate and less uncertain these models become. A higher density of age determination along the depth of the sediment core is desirable for future drilling campaigns (cf. Blaauw et al., 2018).

The disadvantage arises in our second case study *("Inconsistent sequence"* – CS2) and the multi-core investigation *("Multiple cores"* – CS3). For both cases we needed the optimization step to narrow down the most suitable age-depth models for each sediment core, since the unoptimized uncertainty band was otherwise too wide for a clear interpretation. The optimization requires additional and independent proxy data, which are not available for some of our cores, especially for sediment cores obtained some decades ago. Our optimizing step is therefore mainly
suitable for recently retrieved and analyzed sediment cores.

In addition to the assessment of age-modeling quality, we also checked the time and effort to conduct dating routines. We saw that *Bacon* had the highest runtime overall in all three case studies of our study design, which we link to our adjustment of the "ssize" parameter from 2000 (per default) to 8000 within the application. We increased this value to ensure good MCMC mixing for problematic cores, as suggested by Blaauw et al., (2021),
as well as to guarantee we had enough iterations for our summarizing statistics to compare with other modeling systems. If users decide to reduce the value of "ssize", we implemented an iterative process, which checks whether *Bacon* produced enough iterations. If this is not the case, then LANDO will iteratively rerun the same sediment core with a higher "ssize" to produce 10 000 iterations.

One unique feature of our application is the predominant use of parallelization within the age-depth modeling of
multiple sediment cores. For instance, we used the "Dask" back-end for our sedimentation rate calculation. The advantage over popular Scala-based "Apache Spark" and its Python interface "PySpark" (Zaharia et al., 2016) is that the "Dask" back-end is Python-based and well integrated into the Python ecosystem (cf. Dask Development Team, 2016). Therefore, "Dask" natively works with Python packages already implemented in LANDO. The key difference is that "Dask" does neither provide a query optimizer nor rely on Map-Shuffle-Reduce, a data processing
technique for distributed computing, but instead uses a generic task scheduling (cf. Dask Development Team, 2016). Still, parallelization libraries and back-ends provide LANDO with additional speed-up that can promote future multi-core studies.

Within the ensemble model, we faced the challenge that the combination of all age distributions from the underlying age-depth models per centimeter represents a multi-modal distribution, especially in cases such as the
*"Inconsistent sequence"* case study (CS2). It also means that the output of the ensemble model in these cases is susceptible to inclusion/exclusion of any model. However, we consider using the weighted average median age to be a suitable solution for the multi-model distribution problem, as it is a good indicator on the most probable age within each centimeter based on all modeling systems. But we advise users to use the age confidence intervals per centimeter in subsequent analyses, instead of relying solely on the weighted average median age (cf. Telford et al.,
2004). By optimizing the ensemble model with the ability to include independent proxy data, users can increase the likelihood of a more probable mean age for their sediment core.

**4.3 Technical specifications of LANDO**

In the further course of development, we decided to limit the resolution of the age-depth relationships. Using a resolution of one-centimeter increments allows us to match most proxy measurements from each sediment core

with our age-depth models, apart from high-resolution measurement, such as XRF measurements. To allow a matching with high-resolution proxy data, we tested for a higher resolution of 0.25 cm for our application. In the single sediment core cases (CS1 and CS2), this change did not affect the workflow of LANDO. In turn, the *"Multiple cores"* case (CS3) ran into memory issues. Since the SoS notebook and our parallel back-ends store the result data frames in memory, expanding the resulting data frames to a 0.25 cm resolution causes a fourfold increase in memory use, which limits our capability to run our application on a single laptop. As an intermediate solution, we stored the results from each parallelization worker on disk to free the memory and performed combining operations later. Based on this experience, we recommend working with data centers or increasing the available main memory (RAM) of the operating computer for multi-core studies with expected high-resolution output.

Another advantage of parallelization is that most modeling systems only run on one CPU/thread. Nowadays, however, both personal computers and data centers are made up of multiple CPUs/threads. Especially for larger multi-site studies, our application has the advantage of cutting the overall computing time by running each modeling system on multiple CPUs/threads simultaneously, even for personal computers. In comparison to serial execution of multiple models on one CPU/thread, which would take several hours, our parallel execution reduced the computing time per modeling system by a factor up to four. When considering that our setup consisted of six CPUs (12 threads) and 16 GB RAM, user can even further increase this factor by using larger computing facilities.

Sediment core length is the most limiting factor that determines the overall computing time in our application. However, we want to ensure that users can model each sediment core over its entire length to match proxy data with the correct age-depth relationships. Within our LANDO system, we faced this problem by using extrapolation to calculate ages beyond available dating points. The exception here is the modeling system *Undatable*, which models only between the first and last dating point, as these two dating points act as anchors for the bootstrapping process (Lougheed and Obrochta, 2019). As a result, we saw the sedimentation rate dropping twice to zero at the end of the sedimentation rate calculations. We link this behavior to the end of the individual modeling processes of *Undatable* as well as the other implemented systems.

Extrapolating the age-depth models beyond age determination points always bares the risk that the extrapolated dates do not reflect the actual age. The implemented modeling systems account for this circumstance by increasing the uncertainty for these undated regions (Blaauw, 2010). While we are aware of this potential issue, we wanted to allow users to take advantage of the full age-depth coverage for their sediment core. Blaauw et al. (2018) pointed out in their findings that "most existing late-Quaternary studies contain fewer than one date per millennium" and recommended to increase the number of dating points to "a minimum of 2 dates per millennium". This recommendation would further decrease the need of extrapolation and reduce the overall uncertainty of age-depth models. We agree that more age control can improve the age-depth modeling results, but until the associated costs to analyze organic material for radiocarbon dating do not decrease more significantly (Hajdas et al., 2021; Zander et al., 2020), we recommend LANDO as tool to improve age-depth modeling.

**4.4 Current and future model implementation in LANDO**

During the development of our approach, we realized that some programs were not executable or parallelizable under the current circumstances. For instance, we tested *OxCal* 4.4 as stand-alone version on Windows with NodeJS (version 12.13.1.0) and the R package "oxcAAR" (Martin et al., 2021) within our application. In the case

of EN18208, execution duration was above 3 hours until the notebook lost connection to the *OxCal* interface. Furthermore, some cores never fully reached convergence within *OxCal*. We tried adapting our set-ups including changing the internal constraints, i.e. placement and number of boundaries, or using different depositions models, i.e. alternating between sequential model ("Sequence()") and Poisson-process deposition model ("P_Sequence()"). According to Bronk Ramsey and Lee (2013), the long-term plan of *OxCal* is to make the entire source code openly accessible, which we fully support. An open source code would allow us to identify the current bottleneck so that we could implement *OxCal* in a future release.

To determine the most fitting age-depth model through the *clam* modeling software, we added the "best fit" option to LANDO by default. The "best fit" option utilizes the negative log fit results from all *clam* outputs and identifies the fit with the lowest result as best fit. We included two further exclusion criteria for *clam* models within LANDO: if a) there are too many age reversals within the models, or b) the fit reaches infinity. Under specific circumstances, some sediment cores will not have a fitting model, as is the case, for instance, in the *"Inconsistent sequence"* case study (CS2). Including models that do not fit the data would lead to erroneous estimations of the age-depth relationship. This comes with the cost of losing an established model in the combined model, if no fitting *clam* model is available. However, we think that the benefit of having a more fitting model outweighs this cost.

Although *Undatable* is open source and the fastest modeling system within LANDO, its original development environment (MATLAB) is not free of charge. That is why we implemented *Undatable* in the open source MATLAB-equivalent Octave. Since the Octave version of *Undatable* was slower than the original MATLAB version, we used the parallelization package "parallel" (Fujiwara et al., 2021) to provide comparable results in terms of computing time. To use *Undatable* with MATLAB within our application, users must acquire a license of MATLAB and link the MATLAB kernel to their license. Unfortunately, we do not have the capacity to provide individual licenses with LANDO. For users with an active MATLAB license, we provide in the repository mentioned in the "Code and data availability" section the appropriate code to run the MATLAB version of *Undatable* in LANDO.

We highly appreciate all the work that went into developing the stand-alone versions of each modeling system. Because LANDO relies on the work of these modeling systems, we encourage users of LANDO to cite the original modeling software alongside the LANDO publication in their work. Additionally, users should try the stand-alone versions for each modeling system to provide feedback to both LANDO and modeling system maintainers.

A potential expansion option of LANDO within the multi-language environment is to extent the application and allow future data analysis to use powerful tools, such as Python's machine learning libraries, e.g., keras (Chollet and and others, 2015) and tensorflow (Abadi et al., 2016). We anticipate that other developers can use LANDO as their starting point in building larger limnological data analysis application.

## 5   Conclusion

This paper introduced our application LANDO – a linked age-depth modeling notebook approach. We presented an improved age-depth modeling procedure for sediment cores from high-latitude lake systems by linking five established systems: *Bacon, Bchron, clam, hamstr,* and *Undatable*. The added value of our application is the reduced effort to use established modeling systems in a single Jupyter Notebook for both single and multiple dating series and at the same time make the results comparable. In addition, we introduced an ensemble model that uses

the output from all models to create a more robust age-depth relationship. In the case of scattered age determination data, we further implemented an adapted version of the fuzzy change point approach that allow users to integrate independent proxy data as indicator of lithological changes. This option helps evaluate the performance of modeling systems across lithological boundaries while providing a more reliable ensemble age-depth model by filtering inappropriate model runs for problematic datasets. Our application also allows users to run large datasets with multiple sediment cores in parallel to reduce the overall computation time. In our data collection of 55 sediment cores from northern lake systems at high latitudes, we found that the main regime changes in sedimentation rates do not occur synchronously for all lakes at the Pleistocene-Holocene boundary. However, we linked this behavior to the uncertainty within the modeling process as well as the local variability of the sediment cores within the collection.

## Code and data availability

The LANDO code is accessible at GitHub (https://github.com/GPawi/LANDO) (Pfalz, 2022). We provide five example spreadsheets in the repository for users to test the application. A stand-alone version of the LANDO application will be available upon publication. The dataset with all dating points used in this study, including their references, will be accessible via Pangaea.

## Competing interests

The authors declare that they have no conflict of interest.

## Author contribution

GP wrote the manuscript with inputs from all co-authors. GP developed the application, designed and implemented the LANDO system, and conducted testing. BKB, BD, and JCF advised and supervised the work of GP. BKB, BD, LS, and DAS provided published and unpublished age determination data for this publication.

## Acknowledgements

The authors acknowledge the support of the Helmholtz Einstein Berlin International Berlin Research School in Data Science (HEIBRiDS), the Alfred Wegener Institute - Helmholtz Centre for Polar and Marine Research, the Einstein Center Digital Future, the Humboldt University of Berlin, and the Ministry of Education of the Russian Federation as part of a state task (project no. FSZN-2020-0016). We would like to thank the two reviewers Bryan C. Lougheed and Timothy J. Heaton as well as the authors of the community comment, who provided extensive and engaged feedback that helped us improve the manuscript. The authors highly appreciate all the work that went into developing the stand-alone versions of each modeling system. Hence, we like to thank Maarten Blaauw, Andrew C. Parnell, Andrew Dolman, Bryan C. Lougheed, and Stephen P. Obrochta for their continuous work on *Bacon, Bchron, clam, hamstr,* and *Undatable*.

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
