# Peer review of "Improving age-depth relationships by using the LANDO model ensemble"

_Geochronology, 2021_

## Referee Comment (RC1)

In this study, Pfalz et al. present a very useful piece of software (LANDO) that serves as a universal wrapper for simultaneously applying multiple age-depth software packages to age-depth data points. Considering that different age-depth modelling packages each have their own unique approach, it is generally a good idea that researchers consider multiple age-depth modelling software packages to further understand the various software/methods and to see how choice of method might affect interpretation and why. In this respect, LANDO is a very valuable contribution to the research landscape, because it centralises the running of separate software packages into one interface. It is great that the scripts have been made open access on Github. I also think it is great that the authors took the time to make an accessible piece of software with clear installation instructions for Mac and Windows. However, seeing as I use Linux, I could not follow these instructions for installation. The manuscript does provide a good overview of what LANDO does, so I simply review the scientific parts here, as opposed to the LANDO software itself. Finally, I'm also impressed that the authors managed to get *Undatable* running at speed in Octave!

In general, I think the manuscript is very suitable for the journal *Geochronology* and will be of great interest to the readership. In my opinion, some work and clarification can further improve certain parts of the manuscript, which I detail below.

**Main points**

The main issue with the manuscript as it currently stands, in my opinion, is not related to the software itself or how it is described, but how the manuscript uses the LANDO software in an exercise in interpreting the performance of the various age-depth modelling software packages. When comparing the different packages, the authors state:

> *"To lower our impact and to avoid introducing biases in the modeling process, we used the default values from each modeling system as our own default values (Blaauw et al., 2021; Blaauw, 2021; Parnell et al., 2008; Dolman, 2021; Lougheed and Obrochta, 2019)."*

The above highlights the general issue with the parts of manuscript that compare age-depth modelling software packages. All of the age-depth model software packages in the manuscript are compared using "default" settings, but all of the packages have settings for a reason, namely that they should be adjusted. So it is possible that the age-depth software packages are not compared on their merits. I note that the LANDO software has the option to adjust the settings for each software package, so I am not describing a limitation of LANDO here.

I can give an example about how using "default settings" can affect the interpretation in the case of *Undatable*. Figure 3 in the Pfalz et al. manuscript suggests that Undatable exclusively follows the younger dates between 200 and 600 cm, and the authors mention something similar in the manuscript in lines 408 to 410 of their manuscript.

While it is true that the GUI version of Undatable displays some settings in the data entry windows when the GUI first boots up, these are by no means "default values", but rather starting/dummy values in the GUI. The *Undatable* paper (Lougheed and Obrochta, 2019) discusses that bootpc (bootstrapping percentage) should be increased in the case of large age-depth scatter or age-reversals. Indeed, dealing with scatter in this way is stated in Lougheed and Obrochta (2019) as one of the main advantages of Undatable. Seeing as core EN18208 contains such scatter, I have rerun Undatable using a bootpc of 70 (after Gregor Pfalz kindly shared the input data with me), with the following result, with Pfalz et al. Figure 3 shown for comparison:

| Pfalz et al. Figure 3 | EN18208 with Undatable, 70% bootstrapping |
|---|---|

[Figure]

[Figure]

In the above example, the Undatable uncertainty range expands to take into account the scatter of the dataset, and between 200 and 600 cm the highest probability area shifts more towards the centre of the age-depth scatter. This is the intended philosophy behind the deterministic *Undatable*, namely that the uncertainty range of the age-depth model should increase so that the scatter of the age-depth points is taken into account, i.e. 95% of the age-depth points should feasibly be located within the 95% uncertainty range of the age-depth model.

Other age-depth modelling packages also have their own settings and approaches.

**Other points**

A small point regarding interpreting a lack of age-depth reversals as "undisturbed sediment"... Following bioturbation theory (e.g. Berger and Heath, 1968) when the sediment is fully uniformly mixed throughout the deposition history, downcore multispecimen / bulk samples will produce age-depth points that are in chronological order, i.e. lacking age-depth reversals. In other words, a lack of age-depth scatter is not an indicator for undisturbed sediment (despite perhaps 90+% of the literature assuming otherwise).

When describing the performance of the age-depth models, the text describes the age-depth models from top down, whereas most of the algorithms operate in the direction of sedimentation/time, i.e. from bottom up.

In the age-depth model figures, calibrated dates are indicated by black dots with error bars. Please add some information in the legend or caption about what the black dot is (median or mean calibrated age?), and the error bars (+/- 1sigma, i.e. symmetrical error bars, or the central 68% range, i.e. asymmetrical error bars).

The "optimised" age-depth model in Fig 4c takes what can be described as a middle route through the age-depth points, but with very small confidence intervals. It could be argued that such small confidence intervals mask the scatter of the age-depth determinations, and therefore the true geochronological uncertainty. This is more of a philosophical point, however, seeing as some age-depth packages try to find an optimised route between age-depth points with minimal age model uncertainty (e.g. Bacon, Bchron, OxCal), whereas others also expand uncertainty to take into account the scatter in age-depth points (e.g. Undatable). An argument can be made for either

approach, but in a manuscript that compares all the different types of approaches, it would be useful to point them out.

Thanks for providing this interesting manuscript to review, I look forward to seeing the finished product!

Kind regards,
Bryan Lougheed

P.S. Looking forward to seeing LANDO operating in the cloud (city).

---

## Referee Comment (RC3)

**Review of "Improving age-depth correlations by using the LANDO model ensemble"**

Timothy J Heaton

January 12, 2022

**1 Paper Summary**

The paper describes a front-end Jupyter Notebook which allows users to fit multiple different age-depth model (BChron, BACON, clam, hamstr, and undatable). Currently these models are all only available in individual packages (I believe four in $R$ — BChron, BACON, clam and hamstr; and one in MATLAB — undatable). This separation of the age-modelling approaches into individual packages has hindered the community from testing, using and comparing the various age-depth methods.

The aim of the work is also to allow the user to perform model averaging whereby the output of the various age-models can be combined. This is based upon the idea that one should investigate the sensitivity of the output to the specific modelling assumptions which underlie each age-depth approach — none of these assumptions are likely to be entirely correct and so one should see how much difference they make. By creating an ensemble, one can investigate the spread between the various models. While all the models are wrong, hopefully by considering lots of them you can understand how much of a difference the various modelling assumptions make. From that one might aim to get an idea of the further uncertainty due to age-depth model dependence. If all the methods show similar results this boosts confidence, if they do not one should recognise this model dependence.

The work is a valuable contribution to the community who I am sure will benefit from being able to fit multiple models through the same front end. I do however have some points regarding the presentation and case studies which I think need addressing.

**2 Major Conceptual Comments:**

**2.1 Suitable Types of Data**

Implicitly it seems as though when entering $^{14}$C determinations, the data are calibrated against the IntCal20 atmospheric calibration curve — possibly with the application of a reservoir age (although more on that a bit later). Calibration against this IntCal curve is only appropriate for NH atmospheric samples, or for lakes where the reservoir offset is independent of ocean circulation (in such cases the surface water depletion occurs as a potential consequence of both the release of old, but not necessarily dead, organic carbon from soils and peats; and dead inorganic carbon, a hard water effect, entering the lake from its inflows/groundwater). For data from open oceans, one must use the Marine20 calibration curve — it is not appropriate to apply a constant reservoir age to $^{14}$C samples from the open oceans since the open ocean environment considerably smooths/filters the (radiocarbon vs calendar age) variations seen in atmospheric signal. Such smoothing does not occur with the application of a constant reservoir age.

However, there are many applications beyond simply lake sediments where one might wish to use age-depth modelling. Age-depth models are frequently used in ocean sediment cores and in archaeological sites. This broadens the potential scope of LANDO. While the introduction discusses only lake sediments, it is not explained as to when the internal calibration process is appropriate and when it is not. Some explanation is needed here as it is likely users will come across LANDO in other contexts beyond simply lake sediments. Further, permittting the users to select the marine calibration curve (with an appropriate $\Delta R$) would increase the applicability of the tool.

I also note that potentially some of the cores, shown in Figure 1 and used in the third case study, look like they might be more general open ocean cores than solely lake sediment. Is this the case? In which case they really should be calibrated using the Marine20 curve — this may also have an affect on the sedimentation rate estimates around the Holocene-Pleistocene boundary since the (open ocean) marine reservoir age is known to change between these two periods - see Figure 4 and 7 in the Marine20 paper.

**2.2 Reservoir Ages**

The way that reservoir age is applied in LANDO seems to use a different definition of reservoir age to that commonly in use within the $^{14}$C community. For standard IntCal/MarineCal radiocarbon calibration, the reservoir age (at calendar age $\theta$ cal yr BP) is defined as the difference between the radiocarbon age of dissolved inorganic carbon (DIC) in the mixed surface layer of the water at that location, and the radiocarbon age of $CO_2$ in the Northern Hemispheric (NH) atmosphere. In other words, for IntCal and calibration, the reservoir age is measured in $^{14}$C yrs and is applied to the $^{14}$C determination before calibration.

In this paper however, it seems that the LANDO reservoir age is defined as the difference between the calendar age obtained by calibrating directly against the IntCal curve without any adjustment, and the true calendar age. In the preparation step you use the difference in calendar ages between the unadjusted model and the top of the core.

Applying a constant offset in the $^{14}$C domain before calibration of a sample is equivalent to an assumption that a constant proportion of the $^{14}$C in that sample arises from inorganic carbon (e.g., the hard water effect). This is not quite true if you simply shift the calendar ages. For old sediment cores (from the pleistocene) or sparsely sampled cores the difference between the approaches may be relatively small — especially for cores that are incredibly long. However I believe this will cause confusion to users.

**2.3 Application on Inconsistent Cores: Example 2**

It is my very strong belief that no one should be trying to fit an automated age-depth model to the data as shown in Figure 3. It is clear there is something highly unusual and unexplained regarding the measurements in this core. I am not sure if this is due to certain techniques disagreeing e.g. OSL being different from $^{14}$C (since the data are plotted in the same colour and I cannot tell which dates are which). The correct approach would be to go back to the measurements and determine what is going on. In my view, suggesting that data as inconsistent as this can be resolved by forcing them through a range of models (which may or may not happen to select all path through the data) is highly dangerous. This will encourage users to do similarly rather than investigate the root cause of such issues. I believe this case study should not be used for this reason.

In such inconsistent sets of data, I imagine which route the models take through the data will be highly dependent upon their initialisation and, in the case of MCMC, are very unlikely to mix (as can be seen by the narrow uncertainty bands on each individual curve). I do not see this as a case of model averaging (where there are fundamentally different modelling assumptions which are all plausible which lead to different results) but rather luck as to where you initialise each method. In this case perhaps it works ok as the methods happen to choose what look like the most extreme paths however I would think this is to a large extent fortuitous rather than by design. None of the methods individually fit the data at all well. I would argue that trying to average over a lot of models which are all individually catastrophically bad fits does not add much strength. I much prefer, and would suggest any user takes, the approach of Vyse et al. to go back and look at the raw data and understand what is happening.

*Aside: When plotting the data in the cores as solid circles (e.g., Fig 2) I would find it helpful to colour code according to the type of dating used (e.g., OSL, $^{14}C$ ...). This would make it much easier to identify and understand outliers/inconsistencies. Currently, the text around lines 405 – 410 are not understandable as a reader does not know which are the OSL and which are the $^{14}C$ dates.*

**2.4 Method Description**

While I appreciate that the paper is about the development of the coding, I think there needs to be a short description of each age-depth model for the user. This should not repeat the original papers but just give an intuitive explanation so the reader is aware what they are applying, and what the specific assumptions of that technique are. Ideally this section should include an informal discussion of the strengths and weaknesses of each modelling approach. With such a section, a user might be able to decide whether certain models are more appropriate than others for their setting.

**2.5 Checking convergence**

I do not know enough about the specific implementation of the age-depth models in their own packages but is there a way of passing information on model fit to the user. In particular, some of the methods rely upon MCMC and one needs to be assured of convergence; while the frequentist approach (undatable??) may also give some measure of model fit. It may be that the underlying code itself (BACON, BChron, clam, ...) does not provide this but is there a way to obtain information in LANDO that the results of the individual models are appropriate/have converged/or fit?

For example in case study 1, I wondered if it was realistic to have a sedimentation rate that varies from 0.002 to 2.486 cm/yr where the raw $^{14}$C dates suggest an inversion? I do not know enough about this location but is it instead possible that some of the models are not fitting well, have been run with inappropriate parameters, or have not converged properly. This is even more so in case study 2 where I would hope that individually all of the methods would tell you that they are not fitting the data well.

I recognise this is more about the underlying code to which you link than the LANDO implementation — so there may be nothing you can do to resolve this.

**3    Suggested Additional Information**

- I would suggest you need to provide the link to the software very clearly and explicitly in the Introduction. This is what most users will want and currently you have to get to the end to find out how to actually access the software.

- I think it worth perhaps adding a clear caveat that it is not appropriate for a user to try all the age-depth models and then simply select the single answer that they like the most in terms of fitting with a particular hypotheses. I know you are not proposing any user do this but a warning might help ensure proper use of LANDO.

- You seem to have missed the opportunity to discuss the practical differences between the various age-depth models in your examples. Can you identify features that always seem to be present for some models? For example, in case study 1, the red and green sedimentation estimates seem to be much more extreme than the other models. Is this a consistent feature? Are there reasons for this?

- I do not understand Section 2.5.3 — this needs to be written much more clearly. Does this relate to the way that the proxies and the Holocene boundary are used to filter unreasonable models?

- **Important: Table 1 seems to be lacking what I would guess is the most important piece of information — the depth of the measurements. The format of the data on github (and column labelling of the .xls spreadsheet also does not quite correspond to that given in Table 1 - although it is pretty self-explanatory how they transfer).**

- I think it worth adding a note that all users of LANDO should also reference the underlying methods (and their papers) not simply LANDO.

**4    More minor comments**

- I find the use of "correlations" in the title an odd choice. Would it not just be better to have a title "Improving age-depth modelling by using the LANDO model ensemble"?

- I agree with one of the comments from CC1 that the parameters chosen for each model are critical. However, I do slightly disagree that this is entirely the responsibility of the user. A reliable method should provide good default values, or ideally an automated method to select good parameters (possibly using e.g. cross-validation??)

- Table1 on pg4 — What is the relevance of thickness? It seems unclear to me what thickness of the sample layer means, where is the actual sample within the specific layer? If it is an average over the entire layer then the methodology will presumably become more complex since the $^{14}$C determination relates to the average over the respective time period rather than a single calendar year.

- Should one also have the option of selecting a specific radiocarbon calibration curve? This seems to be particularly relevant for marine $^{14}$C samples where one might want to use a Marine curve with a $\Delta R$

- Figures lack actual panel labellings such as (a) or (b), e.g., there is no Figure 2a

- Line 367'ish — you need to make clear that the mean/median figures relate only to this section (108 – 133 cm in depth) not the entire core.

- When plotting the data in the cores as solid circles (e..g Figs 2) I would find it helpful to colour code according to the type of dating used (e.g., OSL, $^{14}$C ...). This would make it much easier to identify and understand outliers/inconsistencies. Currently, the text around lines 405 – 410 are not understandable as a reader does not know which are the OSL and which are the $^{14}$C dates.

- Line 534'ish — Please identify and explain this unusual core in the section discussing he modelling rather than in the conclusion. I am presuming this is the observation in Figure 6 that has a latitude of around 75° where *undatable* has a huge difference from Holocene to Pleistocene which is not replicated in the other models.

---

## Author Comment (AC1)

Dear Bryan Lougheed,

Firstly, we would like to thank you very much for taking the time to review in detail our manuscript on age-depth relationships in our LANDO model ensemble. We are very pleased that you acknowledge the scientific significance and implications of our study.

In the attached .pdf response file (supplement), we provide detailed replies to each individual comment and provide our proposed changes and adjustments to the manuscript that we will carry out and show within the revised manuscript version. We have therefore highlighted your comments in black and *italics* and highlighted our responses in blue.

Thank you again for investing your valuable time in helping us improve our manuscript.

On behalf of all the authors,

Gregor Pfalz
* * *
**Main points**

*The main issue with the manuscript as it currently stands, in my opinion, is not related to the software itself or how it is described, but how the manuscript uses the LANDO software in an exercise in interpreting the performance of the various age-depth modelling software packages. When comparing the different packages, the authors state:*

> *"To lower our impact and to avoid introducing biases in the modeling process, we used the default values from each modeling system as our own default values (Blaauw et al., 2021; Blaauw, 2021; Parnell et al., 2008; Dolman, 2021; Lougheed and Obrochta, 2019)."*

*The above highlights the general issue with the parts of manuscript that compare age-depth modelling software packages. All of the age-depth model software packages in the manuscript are compared using "default" settings, but all of the packages have settings for a reason, namely that they should be adjusted. So it is possible that the age-depth software packages are not compared on their merits. I note that the LANDO software has the option to adjust the settings for each software package, so I am not describing a limitation of LANDO here.*

*I can give an example about how using "default settings" can affect the interpretation in the case of Undatable. Figure 3 in the Pfalz et al. manuscript suggests that Undatable exclusively follows the younger dates between 200 and 600 cm, and the authors mention something similar in the manuscript in lines 408 to 410 of their manuscript.*

*While it is true that the GUI version of Undatable displays some settings in the data entry windows when the GUI first boots up, these are by no means "default values", but rather starting/dummy values in the GUI. The Undatable paper (Lougheed and Obrochta, 2019) discusses that bootpc (bootstrapping percentage) should be increased in the case of large age-depth scatter or age reversals. Indeed, dealing with scatter in this way is stated in Lougheed and Obrochta (2019) as one of the main advantages of*

*Undatable. Seeing as core EN18208 contains such scatter, I have rerun Undatable using a bootpc of 70 (after Gregor Pfalz kindly shared the input data with me), with the following result, with Pfalz et al. Figure 3 shown for comparison:*

*Pfalz et al. Figure 3                                  EN18208 with Undatable, 70% bootstrapping*

[Figure]

*In the above example, the Undatable uncertainty range expands to take into account the scatter of the dataset, and between 200 and 600 cm the highest probability area shifts more towards the centre of the age-depth scatter. This is the intended philosophy behind the deterministic Undatable, namely that the uncertainty range of the age-depth model should increase so that the scatter of the age-depth points is taken into account, i.e. 95% of the age-depth points should feasibly be located within the 95% uncertainty range of the age-depth model.*

*Other age-depth modelling packages also have their own settings and approaches.*

We agree that the parameter selection can directly affect the interpretation of each use case. However, in this manuscript, we focus on how well the models perform on specific input data. As you have shown, increasing the uncertainty band for Undatable would certainly allow all dating points to be included, but still, the mean values of the model and higher probability ranges would not match the model developed by Vyse et al. (2020).

[Figure]

Figure of age-depth model by Vyse et al. (2020)

If one considers that Vyse et al. (2020) used Undatable for their age-depth model but removed dating points prior to execution, this shows that the model performance depends primarily on

the input data. But we also agree with Reviewer 2 that a reliable method should give good default values. As we said on Page 22, Lines 572-575 "*But we also wanted to simplify the process for users who do not have in-depth modeling knowledge. By using the default values, we can compare models based on their ability to work with the available data. **On the other hand, we are sure that the developers have set their default values based on systematic testing.***" Finding appropriate parameters for each of the individual modeling systems would require techniques, such as grid search, and would be highly dependent on the given input data.

It is further questionable which of the two approaches (large uncertainty range vs. narrow uncertainty range) the user will want. We agree that a larger uncertainty band is desirable for paleoenvironmental reconstruction, when propagated into proxy interpretations. However, Lacourse & Gajewski (2020) found that "*[a]lthough 84% of the papers in our literature sample showed uncertainties on a plot of the age model, none made explicit use of the uncertainties in their paleoenvironmental reconstructions*". Since we made sure that user in LANDO can change the settings in each of the modeling systems, which you thankfully mentioned, they can examine the impact of each input parameter on their model and input data. Overall, we agree with your comment and recognize that this is a broader topic that potentially feeds further research projects related to data science methods.

**Reference**:

Vyse, S. A., Herzschuh, U., Andreev, A. A., Pestryakova, L. A., Diekmann, B., Armitage, S. J., and Biskaborn, B. K.: Geochemical and sedimentological responses of arctic glacial Lake Ilirney, Chukotka (far east Russia) to palaeoenvironmental change since ~51.8 ka BP, Quat. Sci. Rev., 247, 106607, https://doi.org/10.1016/j.quascirev.2020.106607, 2020.

Lacourse, T. and Gajewski, K.: Current practices in building and reporting age-depth models, Quat. Res., 96, 28– 38, https://doi.org/10.1017/qua.2020.47, 2020.

***Other points***

*A small point regarding interpreting a lack of age-depth reversals as "undisturbed sediment"… Following bioturbation theory (e.g. Berger and Heath, 1968) when the sediment is fully uniformly mixed throughout the deposition history, downcore multispecimen / bulk samples will produce age-depth points that are in chronological order, i.e. lacking age-depth reversals. In other words, a lack of age-depth scatter is not an indicator for undisturbed sediment (despite perhaps 90+% of the literature assuming otherwise).*

Thank you for this comment. As you rightly pointed out, most of the literature suggest that term undisturbed sediment refers to a lack of reversals. We will change the name of the case study from "undisturbed sediment" to "continuously deposited sediment" to ensure that our work also considers for bioturbation theory.

*When describing the performance of the age-depth models, the text describes the age-depth models from top down, whereas most of the algorithms operate in the direction of sedimentation/time, i.e. from bottom up.*

Thank you – we will change the order in how we describe the performance of the age-depth models in the revised version.

*In the age-depth model figures, calibrated dates are indicated by black dots with error bars. Please add some information in the legend or caption about what the black dot is (median or mean calibrated age?), and the error bars (+/- 1sigma, i.e. symmetrical error bars, or the central 68% range, i.e. asymmetrical error bars).*

Thank you for the suggestion. We will update the figure captions to further describe the black dots - for instance in Figure 2: *"Generated output from LANDO for sediment core EN18218 ($^{14}$C data from Vyse et al., 2021). Panel (a) consists of a comparison between age-depth models from all five implemented modeling systems (left plot) and their calculated sedimentation rate (right plot). Colored solid lines indicate both the median age and median sedimentation rate for all models, while shaded areas represent their respective one-sigma and two-sigma ranges in the same colors with decreasing opacities. Panel (b) shows the ensemble age-depth model (left plot) and its sedimentation rate (right plot). The dashed line in panel (b) represents the weighted average age estimates (left plot) and the weighted average sedimentation rates (right plot) for the ensemble model, while the grey area represents the two-sigma uncertainty, i.e., the outermost limits of two-sigma ranges from all models. Both plots on the left of (a) and (b) show the depth below sediment surface on the inverted y-axis as composite depth of the sediment core in centimeter (cm) and the calibrated ages on the x-axis in calibrated years Before Present (cal. yr BP, i.e., before 1950 CE).* **Black circles within (a) and (b) indicate the calibrated $^{14}$C bulk sediment samples with their mean calibrated age using the IntCal20 calibration curve (Reimer et al., 2020), and their one-sigma uncertainty as error bar.** *The plots on the right display the sedimentation rate in centimeter per year (cm/yr, x-axis as log-scale) against the depth below sediment surface as the composite depth of the sediment core in centimeter (cm, inverted y-axis)."*

In addition, we will change the legend of LANDO to use different symbols to indicate which category the dating point belongs to, e.g., "14C terrestrial fossil" dating points as a square, "14C sediment" dating points remain as circle, or "other" dating points as triangles.

*The "optimised" age-depth model in Fig 4c takes what can be described as a middle route through the age-depth points, but with very small confidence intervals. It could be argued that such small confidence intervals mask the scatter of the age-depth determinations, and therefore the true geochronological uncertainty. This is more of a philosophical point, however, seeing as some age-depth packages try to find an optimised route between age-depth points with minimal age model uncertainty (e.g. Bacon, Bchron, OxCal), whereas others also expand uncertainty to take into account the scatter in age-depth points (e.g. Undatable). An argument can be made for either approach, but in a manuscript that compares all the different types of approaches, it would be useful to point them out.*

We agree that there are several ways how geoscientists approach the age-depth modeling process, as we mentioned in the previous answer. In our case of Figure 4c, the optimized output has a similar form to the original age-depth model from Vyse et al., 2020. However, we agree with your suggestion on pointing out the different approaches. We shall add another section to the discussion part of the revised manuscript, to ensure that users can decide which approach they prefer.

**Reference**:
Vyse, S. A., Herzschuh, U., Andreev, A. A., Pestryakova, L. A., Diekmann, B., Armitage, S. J., and Biskaborn, B. K.: Geochemical and sedimentological responses of arctic glacial Lake

Ilirney, Chukotka (far east Russia) to palaeoenvironmental change since ~51.8 ka BP, Quat. Sci. Rev., 247, 106607, https://doi.org/10.1016/j.quascirev.2020.106607, 2020.

*Regarding the title, "correlation" would (in my mind, anyway) refer to a statistical relationship of some kind between age and depth. So perhaps replace "correlations" with "relationship" or "models"?*

Thank you for the suggestion. If possible as part of the submission process of a revised version, we will change the title to "Improving age-depth relationships by using the LANDO model ensemble".

---

## Author Comment (AC2)

*Responses to Raphael Gromig, Grigory Fedorov, Bernd Wagner, Volker Wennrich, and Martin Melles*

Dear colleagues,

We very much appreciate the extra effort you have shown with your helpful community comment. Your feedback ensures that we can provide and maintain a high quality publication.

In the attached .pdf response file (supplement), we provide detailed replies to each individual comment and provide our proposed changes and adjustments to the manuscript that we will carry out and show within the revised manuscript version. We have therefore highlighted your comments in black and *italics* and highlighted our responses in blue.

Thank you once again for taking the time to provide feedback on our manuscript.

On behalf of all the authors,

Gregor Pfalz
* * *
*(i) Data reference, availability and usage*

*The origin of the data used in the third case study in most cases is not visible directly from the manuscript but has to be investigated via a "Code and Data availability" spreadsheet, which can be accessed by an attached GitHub link. This spreadsheet in 41 cases provides links to the open and free data repository PANGAEA or original publications, which not always contain the original data and descriptions of age model developments (see below), and in 33 cases the reader is asked to request unpublished data. Once access to the data is accomplished, it is not clear from the manuscript in its present form, which of the existing age data eventually became used in the third case study (see example below).*

*In our mind the relevant original publications existing have to be cited in the manuscript directly and included in the reference list, the data used in the third case study has to be clarified, and the unpublished data used has to be presented in a table in this paper or at least made freely accessible via an open database.*

Thank you for the valuable comment. We agree that it is relevant for the reader to be aware of the underlying data used for our calculations. Fortunately, since the submission of the manuscript, several unpublished datasets have become available in journals. We found only seven unpublished datasets that we would exclude from the revised version. In addition, for all of our sediment cores, we shall refer to the publications with the originally published data and age-depth model. We also qualitatively compared the LANDO model results with the original published age-depth model version and adjusted our LANDO model where needed, for example in the cases listed below.

Following your suggestion, we shall add a table on data availability within the manuscript containing six columns: "CoreID", "PaleoLake Database ID", "Age-Depth Model Available", "Repository", "Accessible", and "Paper Reference". Furthermore, we shall include the references given in this

table in the main references of the publication. We will remove the spreadsheet from the GitHub repository. Instead, we shall also add a table with all dating points from the references in the supporting material to make it easier for the reader to follow the data.

*(ii) Missing geological context*

*The LANDO-derived sedimentation rates displayed for 39 sediment cores in Figures 5 and S1 suggest continuous sedimentation up to 21 cal ka BP with variable rates. Some of these sedimentation rates are obviously wrong, due to missing consideration of geological evidence. Two examples are given below.*

It is true that some of the sedimentation rates do not reflect the actual sedimentation rates compared to age-depth models derived using geological evidence. For this reason, we already wrote on Page 21, Lines 527-528 *"Even though LANDO can produce age-depth models for multiple sediment cores ("Multiple cores" – CS3), we must assume limitations in the geoscientific validity for some of the results."* Since our approach is purely data-driven, i.e., without geological interpretation, we are aware that *"[…] the results from our combined model might over- or underestimate the true sedimentation rate[s]"* (Page 21, Lines 531-532). Our overall purpose was to make LANDO user-friendly enough to allow users to analyze multiple sediment cores without special customizations.

Thanks to your comment, we see that we need to give LANDO users more flexibility. In the revised version, we will mention in the manuscript that LANDO works best in multi-core mode when users have continuous dating series, i.e., only cores without hiatus. We are already planning another paper / technical note to update LANDO for purposes other than lake sediments. Our goal in the next paper is to accommodate user customizations for single sediment cores (such as hiatus or special calibration curves) within a multi-core collection.

*First, the sedimentation rates derived for core Co1309 from Ladoga Lake are based on age data, which according to the "Code and Data availability" spreadsheet originate from Andreev et al. (2019) and Savelieva et al. (2019). However, Andreev et al. (2019) only present OSL ages between 118 and 80 ka BP, substantially exceeding the age range of interest here. Savelieva et al. (2019) present the radiocarbon and OSL ages available from the postglacial part of the record, but mention that the age-depth model used originates from Gromig et al. (2019, in Boreas, 48: 330-348), a paper not cited in the manuscript. Gromig et al. (2019) excluded some of the radiocarbon and OSL ages and, on the other hand, added additional age control from varve chronology and correlation with a radiocarbon-dated record close by. Hence, from the references provided it is unclear, which data finally became used for the LANDO calculations presented. Moreover, both Andreev et al. (2019) and Savelieva et al. (2019) mention that the record contains an obvious hiatus, which spans ca. 14-80 ka BP and is described in detail by Gromig et al. (2019). This hiatus is ignored by the LANDO calculations presented, leading to false data at least for the period 21 - 14 ka BP.*

It is correct that we used dating points derived from Andreev et al. (2019) and Savelieva et al. (2019) for Co1309 (Lake Ladoga). **Table 1** represents the input data of Co1309 to LANDO in our original version of the manuscript and **Figure 1** shows the resulting output.

*Table 1 – Input data of Co1309 based on Andreev et al. (2019) and Savelieva et al. (2019). LANDO input parameters "Lab-Location", "Weight", "Pretreatment", "Reservoir Age", and "Reservoir Error" not included in this table for readability.*

| MeasurementID | Thickness (cm) | LabID | Category | Material | Uncalibrated Age (yr BP) | Uncalibrated Age Error (+/- yr) |
|---|---|---|---|---|---|---|
| Co1309 35 | 1 | Col4057 | 14C terrestrial fossil | organic macro remains | 2470 | 54 |
| Co1309 126.8 | 1 | Col4061.1.1 | 14C terrestrial fossil | organic macro remains | 6681 | 58 |
| Co1309 130 | 10 | C-L3832 | other | quartz silt fraction | 7000 | 300 |
| Co1309 131.8 | 1 | Col4065.1.1 | 14C terrestrial fossil | organic macro remains | 10921 | 68 |
| Co1309 152.8 | 1 | Col4062.1.1 | 14C terrestrial fossil | organic macro remains | 12214 | 69 |
| Co1309 533 | 10 | C-L3835 | other | quartz silt fraction | 14000 | 900 |
| Co1309 743 | 10 | C-L3836 | other | quartz silt fraction | 17300 | 800 |
| Co1309 960 | 10 | C-L3837 | other | quartz silt fraction | 21800 | 1100 |
| Co1309 1166 | 10 | C-L3838 | other | quartz silt fraction | 23400 | 1400 |
| Co1309 1403 | 10 | C-L3839 | other | quartz silt fraction | 82200 | 7800 |
| Co1309 1775 | 10 | C-L3841 | other | quartz silt fraction | 90300 | 5300 |
| Co1309 1977 | 10 | C-L3842 | other | quartz silt fraction | 112800 | 4900 |
| Co1309 2160 | 10 | C-L3843 | other | quartz silt fraction | 117600 | 12600 |

[Figure]

***Figure 1*** *– Previous (first submission version) LANDO age-depth model from Co1309 based on input data by Andreev et al. (2019) and Savelieva et al. (2019)*

We apologize for not including the absolute years of the varve count and the radiocarbon date from Lake Pastorskoye (Subetto et al., 2002), used as an anchor point, published by Gromig et al. (2019) in our original version of the manuscript. As we have decided to streamline our data availability section, i.e., one publication per sediment core, we will now only include data reported by Gromig et al. (2019). Gromig et al. (2019) was the only one out of the three publications that provided a complete age-depth model. However, to allow for a comparison between LANDO and the published age-depth model, instead of modeling the entire core length of 22.7 m, we will also stop at the last varve point at 13.23 m. This avoids the problem of extrapolation.

**Table 2** shows all dating points published by Gromig et al. (2019), which used for the new age-depth model of Co1309. By including the new 30 age controls LANDO generates the output in **Figure 2.**

***Table 2*** *– Input data of Co1309 based on Gromig et al. (2019). LANDO input parameters "Lab-Location", "Weight", "Pretreatment", "Reservoir Age", and "Reservoir Error" not included in this table for readability.*

| MeasurementID | Thickness (cm) | LabID | Category | Material | Uncali-brated Age (yr BP) | Uncali-brated Age Error (+/- yr) |
|---|---|---|---|---|---|---|
| Co1309 35 | 1 | Col4057 | 14C terrestrial fossil | organic macro remains | 2470 | 54 |

| | | | | | | |
|---|---|---|---|---|---|---|
| Co 1309 126.8 | 1 | Col4061.1.1 | 14C terrestrial fossil | organic macro remains | 6681 | 58 |
| Co 1309 130 | 10 | C-L3832 | other | quartz silt fraction | 7000 | 300 |
| Co 1309 131.8 | 1 | Col4065.1.1 | 14C terrestrial fossil | organic macro remains | 10921 | 68 |
| Co 1309 152.8 | 1 | Col4062.1.1 | 14C terrestrial fossil | organic macro remains | 12214 | 69 |
| Co 1309 202.7 | 1 | Varve1 | other | varve | 11380 | 140 |
| Co 1309 231.1 | 1 | Varve2 | other | varve | 11480 | 140 |
| Co 1309 268.8 | 1 | Varve3 | other | varve | 11580 | 140 |
| Co 1309 310.7 | 1 | Varve4 | other | varve | 11680 | 140 |
| Co 1309 351.4 | 1 | Varve5 | other | varve | 11780 | 140 |
| Co 1309 386.6 | 1 | Varve6 | other | varve | 11980 | 140 |
| Co 1309 426.9 | 1 | Varve7 | other | varve | 12080 | 140 |
| Co 1309 449.5 | 1 | Varve8 | other | varve | 12180 | 140 |
| Co 1309 472.2 | 1 | Varve9 | other | varve | 12280 | 140 |
| Co 1309 489.3 | 1 | Varve10 | other | varve | 12380 | 140 |
| Co 1309 506 | 1 | Varve11 | other | varve | 12480 | 140 |
| Co 1309 533 | 10 | C-L3835 | other | quartz silt fraction | 14000 | 900 |
| Co 1309 538.6 | 1 | Varve12 | other | varve | 12580 | 140 |
| Co 1309 575.9 | 1 | Varve13 | other | varve | 12680 | 140 |
| Co 1309 581 | 2 | Ua-14803 | 14C terrestrial fossil | Mosses from Lake Pastorskoye | 10745 | 95 |
| Co 1309 616.5 | 1 | Varve14 | other | varve | 12780 | 140 |
| Co 1309 651.7 | 1 | Varve15 | other | varve | 12880 | 140 |
| Co 1309 684.6 | 1 | Varve16 | other | varve | 12980 | 140 |
| Co 1309 724.6 | 1 | Varve17 | other | varve | 13080 | 140 |
| Co 1309 743 | 10 | C-L3836 | other | quartz silt fraction | 17300 | 800 |
| Co 1309 763.6 | 1 | Varve18 | other | varve | 13180 | 140 |
| Co 1309 798.5 | 1 | Varve19 | other | varve | 13280 | 140 |
| Co 1309 846.2 | 1 | Varve20 | other | varve | 13380 | 140 |
| Co 1309 897.4 | 1 | Varve21 | other | varve | 13480 | 140 |
| Co 1309 958.5 | 1 | Varve22 | other | varve | 13580 | 140 |
| Co 1309 960 | 10 | C-L3837 | other | quartz silt fraction | 21800 | 1100 |

| Co1309 1023.6 | 1 | Varve23 | other | varve | 13680 | 140 |
|---|---|---|---|---|---|---|
| Co1309 1107.1 | 1 | Varve24 | other | varve | 13780 | 140 |
| Co1309 1166 | 10 | C-L3838 | other | quartz silt fraction | 23400 | 1400 |
| Co1309 1191 | 1 | Varve25 | other | varve | 13880 | 140 |
| Co1309 1283.5 | 1 | Varve26 | other | varve | 13980 | 140 |
| Co1309 1301.3 | 1 | Varve27 | other | varve | 13894 | 140 |
| Co1309 1315.7 | 1 | Varve28 | other | varve | 13905 | 140 |
| Co1309 1322.4 | 1 | Varve29 | other | varve | 13910 | 140 |

[Figure]

*Figure 4* – *Comparison of age-depth model from Co1309 – Left: original published age-depth model by Gromig et al. (2019), Right: Corrected LANDO output for Co1309 to include in the revised version.*

**Figure 2** shows that LANDO can reproduce the overall age-depth model by Gromig et al. (2019) without removing dating points.

*Second, the sedimentation rates presented for core PG1205 from Basalt Lake in East Greenland are based on radiocarbon ages originally published by Wagner et al. (2000 in Palaeo3, 160: 45-68), although reference is made to the PhD thesis of Wagner (2000). The LANDO calculations suggest continuous and relatively constant sedimentation since at least 21 cal. ka BP. However, both Wagner et al. (2000) and Wagner (2000) state that the lake record consists of a till at its base, which in all likelyhood was deposited during the Milne Land stade 11.30 - 11.15 cal. ka BP, overlaid by ca. 6.4 m of glaciolacustrine sediments deposited with high sedimentation rates during deglaciation and ca. 2.6 m of hemipelagic sediments deposited with much lower rates during the past ca. 10 ka BP. Hence, the calculations conducted by Pfalz et al. obviously neglect the regional glacial history presented and discussed by Wagner et al. (2000) and Wagner (2000) as well as many papers published before and afterwards, giving the wrong impression that this part of East Greenland became deglaciated already prior to 21 cal. ka BP.*

For PG1205 two datasets exist on Pangaea – one dataset (https://doi.pangaea.de/10.1594/PANGAEA.734962) referencing the publication of Wagner et al. (2000) in Palaeo3, one dataset (https://doi.pangaea.de/10.1594/PANGAEA.385643) referencing the modified version of the PhD thesis of Wagner (2000) published in Reports on Polar Research ("Berichte zur Polarforschung"). Both datasets on Pangaea are identical in content. We referenced the dataset from the publication in Reports on Polar Research (https://epic.awi.de/id/eprint/26538/) because this publication was freely available and allowed us to review the content, while the publication in Palaeo3 (https://doi.org/10.1016/S0031-0182(00)00046-8) was initially behind a paywall and therefore not immediately accessible.

To ensure that we do not include grey literature in the references, we now use Wagner et al., (2000) as reference for the core PG1205. For this 9.85 m-long core we list all the publicly available dating points in **Table 3**, which produced the output in **Figure 3**.

*Table 3 – Input data of PG1205 based on Wagner et al. (2000). LANDO input parameters "Lab-Location", "Weight", "Pretreatment", "Reservoir Age", and "Reservoir Error" not included in this table for readability.*

| MeasurementID | Thickness (cm) | LabID | Category | Material | Uncali-brated Age (yr BP) | Uncalibrated Age Error (+/- yr) |
|---|---|---|---|---|---|---|
| PG1205 33 | 2 | OxA-7253 | 14C terrestrial fossil | twigs | 845 | 40 |
| PG1205 41 | 2 | OxA-7286 | 14C terrestrial fossil | twigs | 985 | 50 |
| PG1205 89 | 2 | UtC-8453 | 14C terrestrial fossil | leaves, twigs | 3050 | 80 |
| PG1205 124 | 1 | OxA-7254 | 14C terrestrial fossil | mosses | 4175 | 50 |
| PG1205 149 | 2 | UtC-8222 | 14C terrestrial fossil | leaves, twigs | 5433 | 35 |
| PG1205 181 | 2 | OxA-7287 | 14C terrestrial fossil | leaves, twigs | 6455 | 70 |
| PG1205 241 | 2 | UtC-8454 | 14C terrestrial fossil | leaves, twigs | 8960 | 160 |

[Figure]

*Figure 3* – *Previous (first submission version) LANDO age-depth model from PG1205 based on input data by Wagner et al. (2000)*

Figure 3 shows why LANDO produces an output that is inconsistent with geological evidence. Since dating points are only available for the first 2.5 meters, LANDO has to extrapolate the remaining seven meters to cover the entire sediment core, which is an extreme case compared to other sediment cores. In the manuscript on Page 24, Lines 645-648 we stated that *"[e]xtrapolating the age-depth models beyond age determination points always bares the risk that the extrapolated dates do not reflect the actual age. The implemented modeling systems account for this circumstance by increasing the uncertainty for these undated regions (Blaauw, 2010). While we are aware of this potential issue, we wanted to allow users to take advantage of the full age-depth coverage for their sediment core."* Similar to the Lake Ladoga sediment core Co1309, we changed the length of the sediment core to the last dating point to avoid strong extrapolation in the new version (Figure 4). We shall include an additional paragraph in the revised version addressing these extrapolation/hiatus issues of LANDO, as well as listing the CoreIDs where we had to adjust our models, so that readers can track our adjustments.

[Figure]

*Figure 4 – Comparison of age-depth model from PG1205 – Left: original published age-depth model by Wagner et al. (2000), Right: Corrected LANDO output for PG1205 to include in the revised version.*

*These two examples illustrate that neglecting geological evidence for hiatuses or large changes in the rates of deposition can create much larger errors in age-depth models and resulting sedimentation rates than the employment of an age-depth modelling system that may not be ideal for the record investigated. From the two examples it becomes evident to us that the literature existing for all sediment records used in the third case study, not only Co1309 and PG1205, needs to be (re)studied and discussed to assure that the geological evidence provided is considered in the sedimentation rates calculated.*

Thank you for bringing this important matter to our attention. We agree and have re-examined all the sediment records closely by comparing the originally published age-depth models and the LANDO outputs. In addition to the two cases you mentioned, we found two other case, where we had to adjust the output. In all four cases discovered, we will discuss this issue in the revised version of the manuscript with reference to the original publication.

---

## Author Comment (AC3)

Dear Timothy J. Heaton,

First of all, we would like to thank you very much for taking the time to review our manuscript entitled "Improving age-depth correlations by using the LANDO model ensemble" in detail. We appreciate your comments on how to improve the current version of our manuscript and your advice to use LANDO for purposes other than lake sediments.

In the attached .pdf response file (supplement), we provide detailed replies to each individual comment and provide our proposed changes and adjustments to the current manuscript that we will carry out and show within the revised manuscript version. We also point out potential future work beyond the scope of the current version. We have therefore highlighted your comments in black and *italics* and highlighted our responses in blue.

Thank you once again for taking the time to review our study.

On behalf of all the authors,

Gregor Pfalz
* * *
**2 Major Conceptual Comments:**

**2.1 Suitable Types of Data**

*Implicitly it seems as though when entering $^{14}C$ determinations, the data are calibrated against the IntCal20 atmospheric calibration curve - possibly with the application of a reservoir age (although more on that a bit later). Calibration against this IntCal curve is only appropriate for NH atmospheric samples, or for lakes where the reservoir offset is independent of ocean circulation (in such cases the surface water depletion occurs as a potential consequence of both the release of old, but not necessarily dead, organic carbon from soils and peats; and dead inorganic carbon, a hard water effect, entering the lake from its inflows/groundwater). For data from open oceans, one must use the Marine20 calibration curve - it is not appropriate to apply a constant reservoir age to $^{14}C$ samples from the open oceans since the open ocean environment considerably smooths/filters the (radiocarbon vs calendar age) variations seen in atmospheric signal. Such smoothing does not occur with the application of a constant reservoir age.*

*However, there are many applications beyond simply lake sediments where one might wish to use age-depth modelling. Age-depth models are frequently used in ocean sediment cores and in archaeological sites. This broadens the potential scope of LANDO. While the introduction discusses only lake sediments, it is not explained as to when the internal calibration process is appropriate and when it is not. Some explanation is needed here as it is likely users will come across LANDO in other contexts beyond simply lake sediments. Further, permitting the users to select the marine calibration curve (with an appropriate ΔR) would increase the applicability of the tool.*

*I also note that potentially some of the cores, shown in Figure 1 and used in the third case study, look like they might be more general open ocean cores than solely lake sediment. Is this the case? In which case they really should be calibrated using the Marine20 curve - this may also have an effect on the sedimentation rate estimates around the Holocene-Pleistocene boundary since the (open ocean) marine reservoir age is known to change between these two periods - see Figure 4 and 7 in the Marine20 paper.*

We agree that for open ocean core data you need to calibrate the data with the Marine20 calibration curve. However, all sediment cores included in this study are lake sediment cores. We do not have a single marine sediment core. However, it is already possible to calibrate dates in LANDO using the Marine20 calibration curve. As we state on Page 11, Lines 326-328: "*To include age determination data within the plots, LANDO internally calibrates the radiocarbon data with the "BchronCalibrate" function of the Bchron package (Haslett and Parnell, 2008; Parnell et al., 2008) with either the IntCal20 (Reimer et al.,*

*2020) or Marine20 (Heaton et al., 2020) calibration curve.*" If the user enters "14C marine fossil" as "Category" for the input data (please see Table 1, page 4, "material_category"), LANDO automatically applies the Marine20 curve to this age point and uses the "reservoir age" as ΔR.

So far, we have only used the IntCal20 curve since all our sediment cores are in the northern hemisphere. If users intend to use LANDO for purposes other than lake sediment cores, we would need to make it clearer for the user which calibration curves are applicable. However, this is beyond the scope of this article, which focuses on Arctic lake sediment cores. We may address this in a future update to make LANDO accessible to other areas.

**2.2 Reservoir Ages**

*The way that reservoir age is applied in LANDO seems to use a different definition of reservoir age to that commonly in use within the $^{14}C$ community. For standard IntCal/MarineCal radiocarbon calibration, the reservoir age (at calendar age θ cal yr BP) is defined as the difference between the radiocarbon age of dissolved inorganic carbon (DIC) in the mixed surface layer of the water at that location, and the radiocarbon age of $CO_2$ in the Northern Hemispheric (NH) atmosphere. In other words, for IntCal and calibration, the reservoir age is measured in $^{14}C$ yrs and is applied to the $^{14}C$ determination before calibration.*

*In this paper however, it seems that the LANDO reservoir age is defined as the difference between the calendar age obtained by calibrating directly against the IntCal curve without any adjustment, and the true calendar age. In the preparation step you use the difference in calendar ages between the unadjusted model and the top of the core.*

*Applying a constant offset in the $^{14}C$ domain before calibration of a sample is equivalent to an assumption that a constant proportion of the $^{14}C$ in that sample arises from inorganic carbon (e.g., the hard water effect). This is not quite true if you simply shift the calendar ages. For old sediment cores (from the pleistocene) or sparsely sampled cores the difference between the approaches may be relatively small - especially for cores that are incredibly long. However I believe this will cause confusion to users.*

Thank you for your comment. We assume that there is a slight misunderstanding. Regarding the procedure: First, we assume that there is no reservoir age. We use the hamstr modeling system, which internally calibrates the radiocarbon dates with either the IntCal20 (or the Marine20) calibration curve using the "BchronCalibrate" function of the Bchron package. Then we use hamstr to determine the uppermost layer (0 centimeter of the depth within sediment core = depth below sediment surface) based on the calibrated dates. This means that we build an age-depth model with no reservoir age as input. If we detect a difference between predicted model output for the uppermost layer (e.g., 1200 cal. years BP) and the actual age we know from the expedition (e.g., -68 cal. years BP – for an expedition in 2018), then we assume that the difference between the two values is the reservoir age (e.g., 1268 cal. years BP). LANDO then adds the new reservoir age to the input file, which means that for the actual model runs, all modeling systems within LANDO apply the reservoir age to the $^{14}C$ determination before calibration.

We would also like to mention that the reservoir correction is an additional option in LANDO. As we states on Page 5, Lines 133-134 "***In the absence of a known reservoir age or recent surface sample***, *we used available radiocarbon data points and a fast-calculating modeling system to predict the age of the upper most layer within a sediment core.*" If users know the reservoir age and still use the reservoir correction option, then this would either not produce a suggestion from LANDO (as desired and predicted output match) or LANDO would suggest a higher reservoir correction that user can either accept or ignore.

**2.3 Application on Inconsistent Cores: Example 2**

*It is my very strong belief that no one should be trying to fit an automated age-depth model to the data as shown in Figure 3. It is clear there is something highly unusual and unexplained regarding the measurements in this core. I am not sure if this is due to certain techniques disagreeing e.g. OSL being different from $^{14}C$ (since the data are plotted in the same colour and I cannot tell which dates are which). The correct approach would be to go back to the measurements and determine what is going on. In my view, suggesting that data as inconsistent as this can be resolved by forcing them through a range of models (which may or may not happen to select all path through the data) is highly dangerous. This will encourage users to do similarly rather than investigate the root cause of such issues. I believe this case study should not be used for this reason.*

*In such inconsistent sets of data, I imagine which route the models take through the data will be highly dependent upon their initialisation and, in the case of MCMC, are very unlikely to mix (as can be seen by the narrow uncertainty bands on each individual curve). I do not see this as a case of model averaging (where there are fundamentally different modelling assumptions which are all plausible which lead to different results) but*

*rather luck as to where you initialise each method. In this case perhaps it works ok as the methods happen to choose what look like the most extreme paths however I would think this is to a large extent fortuitous rather than by design. None of the methods individually fit the data at all well. I would argue that trying to average over a lot of models which are all individually catastrophically bad fits does not add much strength. I much prefer, and would suggest any user takes, the approach of Vyse et al. to go back and look at the raw data and understand what is happening.*

*Aside: When plotting the data in the cores as solid circles (e.g., Fig 2) I would find it helpful to colour code according to the type of dating used (e.g., OSL, $^{14}$C ...). This would make it much easier to identify and understand outliers/inconsistencies. Currently, the text around lines 405 - 410 are not understandable as a reader does not know which are the OSL and which are the $^{14}$C dates.*

Thank you for this comment. We are aware that Figure 3 is not the solution to our case study No. 2. We ran our simulations several times and can therefore assure that the path taken does not depend on the initialization of the MCMC. We argue that the path taken by each individual model depends on the underlying methodology of each age-depth modeling system. The results shown in Figure 3 represent the unoptimized solution for an inconsistent set of data. An inconsistent set of data is challenging for any age-depth modeling system, as we have shown in Figure 3. One option is to remove questionable dating points based on geoscientific knowledge for the specific region/lake. According to Lacourse & Gajewski (2020), 33% of the studies they reviewed rejected one or more 14C ages before modeling, primarily because of age reversals. With LANDO, we present an alternative where we use uninformed models and – in case of scattered age dating points – an adapted fuzzy change point detection method (originally proposed by Holloway et al. (2021)). The results shown in Figure 4 represent the optimized solution to case study No. 2. We agree that it is important for any study to look at the data and examine the reason behind the age scattering. We will therefore add a note to our revised manuscript that users should investigate the scatter in addition to using LANDO. If there are still dates in question, users can remove dating points from the input file and run LANDO again.

And thank you for the additional comment. Yes, we will continue to improve the plot function in LANDO.

**Reference**:
Lacourse, T. and Gajewski, K.: Current practices in building and reporting age-depth models, Quat. Res., 96, 28– 38, https://doi.org/10.1017/qua.2020.47, 2020.

Hollaway, M. J., Henrys, P. A., Killick, R., Leeson, A., and Watkins, J.: Evaluating the ability of numerical models to capture important shifts in environmental time series: A fuzzy change point approach, Environ. Model. Softw., 139, 104993, https://doi.org/10.1016/j.envsoft.2021.104993, 2021.

**2.4 Method Description**

*While I appreciate that the paper is about the development of the coding, I think there needs to be a short description of each age-depth model for the user. This should not repeat the original papers but just give an intuitive explanation so the reader is aware what they are applying, and what the specific assumptions of that technique are. Ideally this section should include an informal discussion of the strengths and weaknesses of each modelling approach. With such a section, a user might be able to decide whether certain models are more appropriate than others for their setting.*

Thank you for this suggestion. We will extend the part about the age-depth models in the method section (Page 3, Lines 103-108) by a brief description about the models.

**2.5 Checking convergence**

*I do not know enough about the specific implementation of the age-depth models in their own packages but is there a way of passing information on model fit to the user. In particular, some of the methods rely upon MCMC and one needs to be assured of convergence; while the frequentist approach (undatable??) may also give some measure of model fit. It may be that the underlying code itself (BACON, BChron, clam, ...) does not provide this but is there a way to obtain information in LANDO that the results of the individual models are appropriate/have converged/or fit?*

*For example in case study 1, I wondered if it was realistic to have a sedimentation rate that varies from 0.002 to 2.486 cm/yr where the raw $^{14}$C dates suggest an inversion? I do not know enough about this location but is it instead possible that some of the models are not fitting well, have been run with inappropriate parameters, or*

*have not converged properly. This is even more so in case study 2 where I would hope that individually all of the methods would tell you that they are not fitting the data well.*

*I recognise this is more about the underlying code to which you link than the LANDO implementation | so there may be nothing you can do to resolve this.*

About convergence: Some of the packages provide convergence information in separate methods within each implementation. However, there is no standardized reporting standard for convergence against which we can compare the models. First, we would have to develop such a reporting standard. Then we can build a separate pipeline within LANDO that tracks these convergence values for each sediment core and each modeling system, as our application should continue to work for multiple sediment cores. As this is more a larger feature request, we may address this in a separate technical note manuscript in GChron and update of LANDO. In the initial version of LANDO, there are two extras settings enforced to ensure that models behave appropriately. For Bacon, for instance, we increased the parameter "ssize" as a precaution to ensure good MCMC mixing, as suggested by Blaauw et al. (2021). In clam, we already use the information on the fit to determine the best model for each core, which clam provides as a direct output. About the example: The values you gave refer to the possible values of the outermost two-sigma ranges when considering all models for the entire core length. Both values do not occur at the same time. At the point of inversion suggested by the calibrates [14]C dates, we see high sedimentation rates and only a few models include the dating point at 114.75 cm in their two-sigma uncertainty range. We will rephrase this in the revised manuscript version.

**Reference**:
Blaauw, M., Christen, J. A., and Aquino Lopez, M. A.: rbacon: Age-Depth Modelling using Bayesian Statistics, https://cran.r-project.org/package=rbacon, 2021.

**3 Suggested Additional Information**

- *I would suggest you need to provide the link to the software very clearly and explicitly in the Introduction. This is what most users will want and currently you have to get to the end to find out how to actually access the software.*

  Thank you for this suggestion. We followed the rules of the GChron manuscript preparation guidelines that said that we must present the link to the code in the code availability section. However, if it is possible, we will also provide one in the introduction.

- *I think it worth perhaps adding a clear caveat that it is not appropriate for a user to try all the age-depth models and then simply select the single answer that they like the most in terms of fitting with a particular hypotheses. I know you are not proposing any user do this but a warning might help ensure proper use of LANDO.*

  Good idea. We will include a warning in the discussion section.

- *You seem to have missed the opportunity to discuss the practical differences between the various age-depth models in your examples. Can you identify features that always seem to be present for some models? For example, in case study 1, the red and green sedimentation estimates seem to be much more extreme than the other models. Is this a consistent feature? Are there reasons for this?*

  We will add a description of the practical differences in the method section about the age-depth models.

- *I do not understand Section 2.5.3 - this needs to be written much more clearly. Does this relate to the way that the proxies and the Holocene boundary are used to filter unreasonable models?*

  We understand that the section might feel a bit out of place. We will move it to a different section number and rename the section to "Further analysis - Sedimentation rate development over time". We wrote this section to describe our further analysis that we performed on the multi-core dataset. This section does not apply to the proxies that we use to filter unreasonable models.

- *Important: Table 1 seems to be lacking what I would guess is the most important piece of information - the depth of the measurements. The format of the data on github (and column labelling of the .xls spreadsheet also does not quite correspond to that given in Table 1 - although it is pretty self-explanatory how they transfer).*

We included the depth information of the measurement in the "measurementID". In Table 1, we define the measurementID as "*Composite key composed of a unique CoreID, a blank space, and* **the depth below sediment surface (mid-point cm) with max. two decimal digits of corresponding analytical age measurement** *- example: "CoreA1 100.5", when users obtained sample of CoreA1 between 100 and 101 cm depth*". The measurementID is an essential key / identifier within our database, which we introduced in Pfalz et al. (2021). However, to accommodate other scientist, we have split the measurementID in the example spreadsheet into its two components: CoreID and Depth mid-point (cm).

**Reference**:
Pfalz, G., Diekmann, B., Freytag, J.-C., and Biskaborn, B. K.: Computers and Geosciences Harmonizing heterogeneous multi-proxy data from lake systems, Comput. Geosci., 153, 11, https://doi.org/10.1016/j.cageo.2021.104791, 2021.

- *I think it worth adding a note that all users of LANDO should also reference the underlying methods (and their papers) not simply LANDO.*

This is a point, which we unfortunately missed. We will include this additional remark in our revised manuscript.

**4 More minor comments**
- *I find the use of "correlations" in the title an odd choice. Would it not just be better to have a title "Improving age-depth modelling by using the LANDO model ensemble"?*

Agreed. Where possible, as part of the submission process for a revised version, we will change the title to "Improving age-depth relationships by using the LANDO model ensemble".

- *I agree with one of the comments from CC1 that the parameters chosen for each model are critical. However, I do slightly disagree that this is entirely the responsibility of the user. A reliable method should provide good default values, or ideally an automated method to select good parameters (possibly using e.g. cross-validation??)*

We agree with both Reviewer No. 1 that parameter selection is crucial and your comment that a reliable method should provide good default values. As we said on Page 22, Lines 572-575 "*But we also wanted to simplify the process for users who do not have in-depth modeling knowledge. By using the default values, we can compare models based on their ability to work with the available data.* **On the other hand, we are sure that the developers have set their default values based on systematic testing.**" Our aim is to compare age-depth modeling systems given solely the input data. Finding appropriate parameters for each of the individual modeling systems would require techniques, such as grid search, and would be highly dependent on the given input data. This would require direct involvement of the developers of all modeling packages to make suggestions to user on the possible parameters for given input data.

- *Table 1 on pg4 - What is the relevance of thickness? It seems unclear to me what thickness of the sample layer means, where is the actual sample within the specific layer? If it is an average over the entire layer then the methodology will presumably become more complex since the $^{14}C$ determination relates to the average over the respective time period rather than a single calendar year.*

Thank you for spotting this error. Thickness actually refers to the thickness of the sample used for age determination, not the layer. "*Thickness of the layer*" should be "*thickness of the sample slice*". In most cases measurements of $^{14}C$ sediment bulk samples require a sediment slice of at least one centimeter. However, we have some data entries consisting of bulk samples from larger slices, presumably to ensure a successful measurement due to the low carbon content within the sediment. Four out of five age-depth

modeling packages within LANDO require thickness as an input parameter. Therefore, we included this parameter in our "necessary" category. The models do not internally average over the entire slice, but instead apply a distribution for each sample. We will correct the description of thickness to "***Thickness of the sample slice used for age determination in [cm]***"

- *Should one also have the option of selecting a specific radiocarbon calibration curve? This seems to be particularly relevant for marine $^{14}$C samples where one might want to use a Marine curve with a $\Delta R$*

As we stated before, it is already possible to calibrate dates within LANDO using the Marine20 calibration curve. In the current manuscript, however, our focus is on terrestrial lakes from the northern hemisphere. In an updated version of LANDO for other use cases, we will include the option for selecting the calibration curve individually.

- *Figures lack actual panel labellings such as (a) or (b), e.g., there is no Figure 2a*

All figures with panels should have a panel label to the right of the sedimentation rate curve. Please find below Figure 2, where we highlighted the location of the label with green circles. We apologize, if you have received a copy our manuscript without these labels.

[Figure]

- *Line 367'ish - you need to make clear that the mean/median figures relate only to this section (108 - 133 cm in depth) not the entire core.*

Thank you for spotting this inconsistency. We will rewrite this sentence to "*All models revealed highest sedimentation rates for the interval between 108 and 133 cm. Mean values ranged from 0.242 cm/yr*

*(hamstr) to 0.764 cm/yr (clam)* **within this interval**, *whereas the median sedimentation rate varied between 0.107 cm/yr (Bacon) and 0.314 cm/yr (clam)."*

- *When plotting the data in the cores as solid circles (e..g Figs 2) I would find it helpful to colour code according to the type of dating used (e.g., OSL, 14C ...). This would make it much easier to identify and understand outliers/inconsistencies. Currently, the text around lines 405-410 are not understandable as a reader does not know which are the OSL and which are the $^{14}C$ dates.*

Thank you for this suggestion. Since we want to reserve colors for possible additions of age-depth models to LANDO, such as OxCal, we will present an alternative solution in the revised manuscript. We will change the legend of LANDO to use different symbols to indicate which category the dating point belongs to, e.g., "14C terrestrial fossil" dating points as a square, "14C sediment" dating points remain as circle, or "other" (such as OSL) dating points as triangles. We hope this will improve the readability of our plots.

- *Line 534'ish | Please identify and explain this unusual core in the section discussing he modelling rather than in the conclusion. I am presuming this is the observation in Figure 6 that has a latitude of around 75° where undatable has a huge difference from Holocene to Pleistocene which is not replicated in the other models.*

We wrote on Page 21, Lines 533-536: *"For instance, during the examination of the Holocene and the Pleistocene sedimentation rates (Figure 6), we noticed that one sediment core (PG1228) had an extremely high mean sedimentation rate for the Holocene dataset in Undatable. Similar to the second case study ("Inconsistent sequence" – CS2), we found scattered age data points for this sediment core, which influenced the modeling process of Undatable."* The paragraph you are referring to is in the discussion section, which discusses all the different scenarios, not in the conclusion section. We identified the core as PG1228, explained that the deviation in sedimentation rate is due to scattered age data points and – similar to case study No. 2 - the different path Undatable takes through the scattered data points. However, due to the community comment, we will revise this graphic with the changed input data and ensure we mention any new observations.

---

## Author Response (AR1)

**Point-by-point changes and responses to comments**

**Main changes**

- Users can now select different calibration curves (IntCal20, Marine20, SHCal20) for their samples
- We reduced the number of sediment cores from 63 to 55, but now only include published datasets
- We compared the LANDO output with previously published age-depth models and adjusted four sediment cores to match the published output
- We changed the legend and plot functionality of LANDO to improve readability

**Referee #1 – Bryan Lougheed**

**_Main points_**

_The main issue with the manuscript as it currently stands, in my opinion, is not related to the software itself or how it is described, but how the manuscript uses the LANDO software in an exercise in interpreting the performance of the various age-depth modelling software packages. When comparing the different packages, the authors state:_

> _"To lower our impact and to avoid introducing biases in the modeling process, we used the default values from each modeling system as our own default values (Blaauw et al., 2021; Blaauw, 2021; Parnell et al., 2008; Dolman, 2021; Lougheed and Obrochta, 2019)."_

_The above highlights the general issue with the parts of manuscript that compare age-depth modelling software packages. All of the age-depth model software packages in the manuscript are compared using "default" settings, but all of the packages have settings for a reason, namely that they should be adjusted. So it is possible that the age-depth software packages are not compared on their merits. I note that the LANDO software has the option to adjust the settings for each software package, so I am not describing a limitation of LANDO here._

_I can give an example about how using "default settings" can affect the interpretation in the case of Undatable. Figure 3 in the Pfalz et al. manuscript suggests that Undatable exclusively follows the younger dates between 200 and 600 cm, and the authors mention something similar in the manuscript in lines 408 to 410 of their manuscript._

_While it is true that the GUI version of Undatable displays some settings in the data entry windows when the GUI first boots up, these are by no means "default values", but rather starting/dummy values in the GUI. The Undatable paper (Lougheed and Obrochta, 2019) discusses that bootpc_

*(bootstrapping percentage) should be increased in the case of large age-depth scatter or age reversals. Indeed, dealing with scatter in this way is stated in Lougheed and Obrochta (2019) as one of the main advantages of Undatable. Seeing as core EN18208 contains such scatter, I have rerun Undatable using a bootpc of 70 (after Gregor Pfalz kindly shared the input data with me), with the following result, with Pfalz et al. Figure 3 shown for comparison:*

*Pfalz et al. Figure 3                              EN18208 with Undatable, 70% bootstrapping*

[Figure]

*In the above example, the Undatable uncertainty range expands to take into account the scatter of the dataset, and between 200 and 600 cm the highest probability area shifts more towards the centre of the age-depth scatter. This is the intended philosophy behind the deterministic Undatable, namely that the uncertainty range of the age-depth model should increase so that the scatter of the age-depth points is taken into account, i.e. 95% of the age-depth points should feasibly be located within the 95% uncertainty range of the age-depth model.*

*Other age-depth modelling packages also have their own settings and approaches.*

As discussed in the interactive discussion, we agree with your comment and recognize that this is a broader topic that potentially feeds further research projects related to data science methods. We additionally added to our manuscript the following sentence (Page 25, Lines 559-561): "*For example, to deal with the scatter in the data, users can increase the Undatable parameter "bootpc" to a higher value - as suggested by Lougheed and Obrochta (2019) - to account for a higher uncertainty in the given data.*"

**Other points**

*A small point regarding interpreting a lack of age-depth reversals as "undisturbed sediment"... Following bioturbation theory (e.g. Berger and Heath, 1968) when the sediment is fully uniformly mixed throughout the deposition history, downcore multispecimen / bulk samples will produce*

*age-depth points that are in chronological order, i.e. lacking age-depth reverals. In other words, a lack of age-depth scatter is not an indicator for undisturbed sediment (despite perhaps 90+% of the literature assuming otherwise).*

Thank you for this comment. We changed the name of the second case study from "Undisturbed sequence" to "Continuously deposited sequence" to ensure that our work also considers for bioturbation theory.

*When describing the performance of the age-depth models, the text describes the age-depth models from top down, whereas most of the algorithms operate in the direction of sedimentation/time, i.e. from bottom up.*

Thank you – we changed the order in how we describe the performance of the age-depth models in the revised version.

*In the age-depth model figures, calibrated dates are indicated by black dots with error bars. Please add some information in the legend or caption about what the black dot is (median or mean calibrated age?), and the error bars (+/- 1sigma, i.e. symmetrical error bars, or the central 68% range, i.e. asymmetrical error bars).*

Thank you for the suggestion. We changed the legend of LANDO to use eight different symbols to indicate which category the dating point belongs to. We have updated the figure captions to match their respective symbols and description of error bars.

*The "optimised" age-depth model in Fig 4c takes what can be described as a middle route through the age-depth points, but with very small confidence intervals. It could be argued that such small confidence intervals mask the scatter of the age-depth determinations, and therefore the true geochronological uncertainty. This is more of a philosophical point, however, seeing as some age-depth packages try to find an optimised route between age-depth points with minimal age model uncertainty (e.g. Bacon, Bchron, OxCal), whereas others also expand uncertainty to take into account the scatter in age-depth points (e.g. Undatable). An argument can be made for either approach, but in a manuscript that compares all the different types of approaches, it would be useful to point them out.*

We agree with your suggestion on pointing out the different approaches. We added a section to the discussion part of the revised manuscript, to ensure that users can decide which approach they prefer (Pages 25-26, Lines 562-569).

*Regarding the title, "correlation" would (in my mind, anyway) refer to a statistical relationship of some kind between age and depth. So perhaps replace "correlations" with "relationship" or "models"?*

Thank you for the suggestion. We have changed the title to "Improving age-depth relationships by using the LANDO model ensemble".

**Referee #2 – Timothy J. Heaton**

*2 Major Conceptual Comments:*

*2.1 Suitable Types of Data*
*Implicitly it seems as though when entering $^{14}$C determinations, the data are calibrated against the IntCal20 atmospheric calibration curve -  possibly with the application of a reservoir age (although more on that a bit later). Calibration against this IntCal curve is only appropriate for NH atmospheric samples, or for lakes where the reservoir offset is independent of ocean circulation (in such cases the surface water depletion occurs as a potential consequence of both the release of old, but not necessarily dead, organic carbon from soils and peats; and dead inorganic carbon, a hard water effect, entering the lake from its inflows/groundwater). For data from open oceans, one must use the Marine20 calibration curve - it is not appropriate to apply a constant reservoir age to $^{14}$C samples from the open oceans since the open ocean environment considerably smooths/filters the (radiocarbon vs calendar age) variations seen in atmospheric signal. Such smoothing does not occur with the application of a constant reservoir age.*
*However, there are many applications beyond simply lake sediments where one might wish to use age-depth modelling. Age-depth models are frequently used in ocean sediment cores and in archaeological sites. This broadens the potential scope of LANDO. While the introduction discusses only lake sediments, it is not explained as to when the internal calibration process is appropriate and when it is not. Some explanation is needed here as it is likely users will come across LANDO in other contexts beyond simply lake sediments. Further, permittting the users to select the marine calibration curve (with an appropriate ΔR) would increase the applicability of the tool.*
*I also note that potentially some of the cores, shown in Figure 1 and used in the third case study, look like they might be more general open ocean cores than solely lake sediment. Is this the case? In which case they really should be calibrated using the Marine20 curve - this may also have an effect on the sedimentation rate estimates around the Holocene-Pleistocene boundary since the (open ocean) marine reservoir age is known to change between these two periods - see Figure 4 and 7 in the Marine20 paper.*

Again, thank you for your valuable comment. To extend our answer during the interactive discussion, it is now possible to use LANDO for purposes other than lake sediment cores. We state on Page 14, Lines 335- 338: "*To include age determination data within the plots, LANDO internally calibrates the radiocarbon data with the "BchronCalibrate" function of the Bchron package (Haslett and Parnell, 2008; Parnell et al., 2008) with either the*

*IntCal20 (Reimer et al., 2020), Marine20 (Heaton et al., 2020), or SHCal20 (Hogg et al., 2020) calibration curve. This allows users to analyze samples from locations other than the terrestrial northern hemisphere."* Before calibration, users can also specify the calibration curve for individual samples instead of using one calibration curve for all samples.

*2.2 Reservoir Ages*
*The way that reservoir age is applied in LANDO seems to use a different definition of reservoir age to that commonly in use within the $^{14}C$ community. For standard IntCal/MarineCal radiocarbon calibration, the reservoir age (at calendar age θ cal yr BP) is defined as the difference between the radiocarbon age of dissolved inorganic carbon (DIC) in the mixed surface layer of the water at that location, and the radiocarbon age of $CO_2$ in the Northern Hemispheric (NH) atmosphere. In other words, for IntCal and calibration, the reservoir age is measured in $^{14}C$ yrs and is applied to the $^{14}C$ determination before calibration.*

*In this paper however, it seems that the LANDO reservoir age is defined as the difference between the calendar age obtained by calibrating directly against the IntCal curve without any adjustment, and the true calendar age. In the preparation step you use the difference in calendar ages between the unadjusted model and the top of the core.*

*Applying a constant offset in the $^{14}C$ domain before calibration of a sample is equivalent to an assumption that a constant proportion of the $^{14}C$ in that sample arises from inorganic carbon (e.g., the hard water effect). This is not quite true if you simply shift the calendar ages. For old sediment cores (from the pleistocene) or sparsely sampled cores the difference between the approaches may be relatively small - especially for cores that are incredibly long. However I believe this will cause confusion to users.*

Thank you for your comment. We assume that there is a slight misunderstanding and hope that our answer in the interactive discussion was sufficient.

*2.3 Application on Inconsistent Cores: Example 2*
*It is my very strong belief that no one should be trying to fit an automated age-depth model to the data as shown in Figure 3. It is clear there is something highly unusual and unexplained regarding the measurements in this core. I am not sure if this is due to certain techniques disagreeing e.g. OSL being different from $^{14}C$ (since the data are plotted in the same colour and I cannot tell which dates are which). The correct approach would be to go back to the measurements and determine what is going on. In my view, suggesting that data as inconsistent as this can be resolved by forcing them through a range of models (which may or may not happen to select all path through the data) is highly dangerous. This will encourage users to do similarly rather than investigate the root cause of such issues. I believe this case study should not be used for this reason.*

*In such inconsistent sets of data, I imagine which route the models take through the data will be highly dependent upon their initialisation and, in the case of MCMC, are very unlikely to mix (as can be seen by the narrow uncertainty bands on each individual curve). I do not see this as a case of model averaging (where there are fundamentally different modelling assumptions which are all plausible which lead to different results) but rather luck as to where you initialise each method. In this case perhaps it works ok as the methods happen to choose what look like the most extreme paths however I would think this is to a large extent fortuitous rather than by design.*

*None of the methods individually fit the data at all well. I would argue that trying to average over a lot of models which are all individually catastrophically bad fits does not add much strength. I much prefer, and would suggest any user takes, the approach of Vyse et al. to go back and look at the raw data and understand what is happening.*

*Aside: When plotting the data in the cores as solid circles (e.g., Fig 2) I would find it helpful to colour code according to the type of dating used (e.g., OSL, $^{14}C$ ...). This would make it much easier to identify and understand outliers/inconsistencies. Currently, the text around lines 405 - 410 are not understandable as a reader does not know which are the OSL and which are the $^{14}C$ dates.*

Thank you for this comment. We agree that it is important for any study to look at the data and examine the reason behind the age scattering. We added the following note to our revised manuscript (Page 25, Lines 557-560): *"Although we chose the highest matching score to demonstrate LANDO's ability of filtering out disagreeing models, we do not support the strategy of choosing a single age-depth model with such a low matching score. Rather, users should investigate the cause of the scatter in the age determination data and/or change the default values within LANDO."* We have further improved the plot function in LANDO. Different symbols now indicate the specific material category of the sample.

*2.4 Method Description*
*While I appreciate that the paper is about the development of the coding, I think there needs to be a short description of each age-depth model for the user. This should not repeat the original papers but just give an intuitive explanation so the reader is aware what they are applying, and what the specific assumptions of that technique are. Ideally this section should include an informal discussion of the strengths and weaknesses of each modelling approach. With such a section, a user might be able to decide whether certain models are more appropriate than others for their setting.*

Thank you for this suggestion. We extended the part about the age-depth models in the method section by a brief description about the models (Pages 3-4, Lines 108-133).

*2.5 Checking convergence*
*I do not know enough about the specific implementation of the age-depth models in their own packages but is there a way of passing information on model fit to the user. In particular, some of the methods rely upon MCMC and one needs to be assured of convergence; while the frequentist approach (undatable??) may also give some measure of model fit. It may be that the underlying code itself (BACON, BChron, clam, ...) does not provide this but is there a way to obtain information in LANDO that the results of the individual models are appropriate/have converged/or fit?*

*For example in case study 1, I wondered if it was realistic to have a sedimentation rate that varies from 0.002 to 2.486 cm/yr where the raw $^{14}C$ dates suggest an inversion? I do not know enough about this location but is it instead possible that some of the models are not fitting well, have been run with inappropriate parameters, or have not converged properly. This is even more*

*so in case study 2 where I would hope that individually all of the methods would tell you that they are not fitting the data well.*

*I recognise this is more about the underlying code to which you link than the LANDO implementation | so there may be nothing you can do to resolve this.*

About convergence: Some of the packages provide convergence information in separate methods within each implementation. However, there is no standardized reporting standard for convergence against which we can compare the models. First, we would have to develop such a reporting standard. Then we can build a separate pipeline within LANDO that tracks these convergence values for each sediment core and each modeling system, as our application should continue to work for multiple sediment cores. As this is more a larger feature request, we will address this in a separate update of LANDO. In the initial version of LANDO, there are two extras settings enforced to ensure that models behave appropriately. For Bacon, for instance, we increased the parameter "ssize" as a precaution to ensure good MCMC mixing, as suggested by Blaauw et al. (2021). In clam, we already use the information on the fit to determine the best model for each core, which clam provides as a direct output.

About the example: The values you gave refer to the possible values of the outermost two-sigma ranges when considering all models for the entire core length. Both values do not occur at the same time. We wrote in the revised version (Page 17, Lines 411-412): *"Throughout the core, the cumulative two-sigma uncertainty of the ensemble model ranged from 0.002 cm/yr to 2.486 cm/yr."* At the point of inversion suggested by the calibrates $^{14}$C dates, we see high sedimentation rates and only a few models include the dating point at 114.75 cm in their two-sigma uncertainty range.

*3 Suggested Additional Information*

- *I would suggest you need to provide the link to the software very clearly and explicitly in the Introduction. This is what most users will want and currently you have to get to the end to find out how to actually access the software.*

  Thank you for this suggestion. In addition to the link in the code availability section (according to the GChron manuscript preparation guidelines), we have provided a link to the main repository in the introduction (Page 3, Lines 87-88).

- *I think it worth perhaps adding a clear caveat that it is not appropriate for a user to try all the age-depth models and then simply select the single answer that they like the most in terms of fitting with a particular hypotheses. I know you are not proposing any user do this but a warning might help ensure proper use of LANDO.*

  Good idea. We included the following warning in the discussion section (Page 25, Lines 556-560).

- *You seem to have missed the opportunity to discuss the practical differences between the various age-depth models in your examples. Can you identify features that always seem to be present for some models? For example, in case study 1, the red and green sedimentation estimates seem to be much more extreme than the other models. Is this a consistent feature? Are there reasons for this?*

  We added a description of the practical differences in the method section about the age-depth models (Pages 3-4, Lines 108-133).

- *I do not understand Section 2.5.3 - this needs to be written much more clearly. Does this relate to the way that the proxies and the Holocene boundary are used to filter unreasonable models?*

  We understand that the section might feel a bit out of place. We moved the mentioned section to section number 2.6 and renamed it to "Further analysis - Sedimentation rate development over time". We wrote this section to describe our further analysis that we performed on the multi-core dataset. This section does not apply to the proxies that we use to filter unreasonable models.

- *Important: Table 1 seems to be lacking what I would guess is the most important piece of information - the depth of the measurements. The format of the data on github (and column labelling of the .xls spreadsheet also does not quite correspond to that given in Table 1 - although it is pretty self-explanatory how they transfer).*

  We included the depth information of the measurement in the "measurementID". In Table 1, we define the measurementID as "*Composite key composed of a unique CoreID, a blank space, and **the depth below sediment surface (mid-point cm) with max. two decimal digits of corresponding analytical age measuremen**t - example: "CoreA1 100.5", when users obtained sample of CoreA1 between 100 and 101 cm depth*".

- *I think it worth adding a note that all users of LANDO should also reference the underlying methods (and their papers) not simply LANDO.*

  This is a point, which we unfortunately missed. We included the following remark in our revised manuscript (Page 30, Lines 742-744): *"We highly appreciate all the work that went into developing the stand-alone versions of each modeling system. Because LANDO relies on the work of these modeling systems, we encourage users of LANDO to cite the original modeling software alongside the LANDO publication in their work."*

*4 More minor comments*
- *I find the use of "correlations" in the title an odd choice. Would it not just be better to have a title "Improving age-depth modelling by using the LANDO model ensemble"?*

Agreed. We changed the title to "Improving age-depth relationships by using the LANDO model ensemble".

- *I agree with one of the comments from CC1 that the parameters chosen for each model are critical. However, I do slightly disagree that this is entirely the responsibility of the user. A reliable method should provide good default values, or ideally an automated method to select good parameters (possibly using e.g. cross-validation??)*

  We agree with both Reviewer No. 1 that parameter selection is crucial and your comment that a reliable method should provide good default values. As we said on Page 27, Lines 632-635 "*But we also wanted to simplify the process for users who do not have in-depth modeling knowledge. By using the default values, we can compare models based on their ability to work with the available data.* **On the other hand, we are sure that the developers have set their default values based on systematic testing.**" Our aim is to compare age-depth modeling systems given solely the input data. Finding appropriate parameters for each of the individual modeling systems would require techniques, such as grid search, and would be highly dependent on the given input data. This would require direct involvement of the developers of all modeling packages to make suggestions to user on the possible parameters for given input data.

- *Table1 on pg4 - What is the relevance of thickness? It seems unclear to me what thickness of the sample layer means, where is the actual sample within the specific layer? If it is an average over the entire layer then the methodology will presumably become more complex since the $^{14}C$ determination relates to the average over the respective time period rather than a single calendar year.*

  Thank you for spotting this error. We changed the description of thickness in Table 1 to "***Thickness of the sample slice used for age determination in [cm]***"

- *Should one also have the option of selecting a specific radiocarbon calibration curve? This seems to be particularly relevant for marine $^{14}C$ samples where one might want to use a Marine curve with a ΔR*

  As we stated before, it is now possible to calibrate dates within LANDO using the different calibration curve (Page 14, Lines 335- 338).

- *Figures lack actual panel labellings such as (a) or (b), e.g., there is no Figure 2a*

  All figures with panels should have a panel label to the right of the sedimentation rate curve. We apologize, if you have received a copy our manuscript without these labels.

- *Line 367'ish - you need to make clear that the mean/median figures relate only to this section (108 - 133 cm in depth) not the entire core.*

Thank you for spotting this inconsistency. We rewrote this sentence to (Page 17, Lines 399-401) *"Mean values ranged from 0.242 cm/yr (hamstr) to 0.764 cm/yr (clam)* **within this interval**, *whereas the median sedimentation rate varied between 0.107 cm/yr (Bacon) and 0.314 cm/yr (clam)."*

- *When plotting the data in the cores as solid circles (e..g Figs 2) I would find it helpful to colour code according to the type of dating used (e.g., OSL, 14C ...). This would make it much easier to identify and understand outliers/inconsistencies. Currently, the text around lines 405 - 410 are not understandable as a reader does not know which are the OSL and which are the $^{14}C$ dates.*

  Thank you for this suggestion. Since we want to reserve colors for possible additions of age-depth models to LANDO, such as OxCal, we changed the legend of LANDO to use different symbols to indicate which category the dating point belongs to. We hope this will improve the readability of our plots.

- *Line 534'ish | Please identify and explain this unusual core in the section discussing he modelling rather than in the conclusion. I am presuming this is the observation in Figure 6 that has a latitude of around 75° where undatable has a huge difference from Holocene to Pleistocene which is not replicated in the other models.*

  As we explained in the interactive discussion, we mentioned this particular core in the discussion section of the manuscript, but it is now on Page 26, Lines 577-580.

**Community comment**

*(i) Data reference, availability and usage*

*The origin of the data used in the third case study in most cases is not visible directly from the manuscript but has to be investigated via a "Code and Data availability" spreadsheet, which can be accessed by an attached GitHub link. This spreadsheet in 41 cases provides links to the open and free data repository PANGAEA or original publications, which not always contain the original data and descriptions of age model developments (see below), and in 33 cases the reader is asked to request unpublished data. Once access to the data is accomplished, it is not clear from the manuscript in its present form, which of the existing age data eventually became used in the third case study (see example below).*

*In our mind the relevant original publications existing have to be cited in the manuscript directly and included in the reference list, the data used in the third case study has to be clarified, and the unpublished data used has to be presented in a table in this paper or at least made freely accessible via an open database.*

Thank you for the valuable comment. We agree that it is relevant for the reader to be aware of the underlying data used for our calculations. Fortunately, since the submission of the manuscript, several unpublished datasets have become available in journals. We found only seven unpublished datasets that we excluded from the revised version. In addition, for all of our sediment cores, we referred to the publications with the originally published data and age-depth model. We also qualitatively compared the LANDO model results with the original published age-depth model version and adjusted our LANDO model where needed, for example in the cases listed below.

Following your suggestion, we added a table (Table 4) on data availability within the manuscript containing six columns: "CoreID", "PaleoLake Database ID", "Age-Depth Model Available", "Main Data Source / Repository", "Data Accessible", and "Paper Reference". Furthermore, we included the references given in this table in the main references of the publication. We removed the spreadsheet from the GitHub repository. Instead, we created a table with all dating points including their original reference and submitted the data to Pangaea. This dataset will soon be available to the public.

*(ii) Missing geological context*

*The LANDO-derived sedimentation rates displayed for 39 sediment cores in Figures 5 and S1 suggest continuous sedimentation up to 21 cal ka BP with variable rates. Some of these sedimentation rates are obviously wrong, due to missing consideration of geological evidence. Two examples are given below.*

It is true that some of the sedimentation rates do not reflect the actual sedimentation rates compared to age-depth models derived using geological evidence. For this reason, we wrote on Page 26, Lines 570-571 "*Even though LANDO can produce age-depth models for multiple sediment cores ("Multiple cores" – CS3), we must assume limitations in the geoscientific validity for some of the results*." Since our approach is purely data-driven, i.e., without geological interpretation, we are aware that "*[…] the results from our combined model might over- or underestimate the true sedimentation rate[s]*" (Page 26, Lines 574-575). Our overall purpose was to make LANDO user-friendly enough to allow users to analyze multiple sediment cores without special customizations.

Thanks to your comment, we added the following remark to the manuscript (Page 26, Lines 598-599): "*However, while a specific customization (such as a hiatus) is possible for single core cases, this is not possible in the current version of LANDO for multi-core investigation.*"

*First, the sedimentation rates derived for core Co1309 from Ladoga Lake are based on age data, which according to the "Code and Data availability" spreadsheet originate from Andreev et al.*

*(2019) and Savelieva et al. (2019). However, Andreev et al. (2019) only present OSL ages between 118 and 80 ka BP, substantially exceeding the age range of interest here. Savelieva et al. (2019) present the radiocarbon and OSL ages available from the postglacial part of the record, but mention that the age-depth model used originates from Gromig et al. (2019, in Boreas, 48: 330-348), a paper not cited in the manuscript. Gromig et al. (2019) excluded some of the radiocarbon and OSL ages and, on the other hand, added additional age control from varve chronology and correlation with a radiocarbon-dated record close by. Hence, from the references provided it is unclear, which data finally became used for the LANDO calculations presented. Moreover, both Andreev et al. (2019) and Savelieva et al. (2019) mention that the record contains an obvious hiatus, which spans ca. 14-80 ka BP and is described in detail by Gromig et al. (2019). This hiatus is ignored by the LANDO calculations presented, leading to false data at least for the period 21 - 14 ka BP.*

> We apologize for not including the absolute years of the varve count and the radiocarbon date from Lake Pastorskoye (Subetto et al., 2002), used as an anchor point, published by Gromig et al. (2019) in our original version of the manuscript. We now included data reported by Gromig et al. (2019). However, to allow for a comparison between LANDO and the published age-depth model, instead of modeling the entire core length of 22.7 m, we stopped at the last varve point at 13.23 m. This avoided the problem of extrapolation. As shown in the interactive discussion, LANDO can reproduce the overall age-depth model by Gromig et al. (2019) without removing dating points.

*Second, the sedimentation rates presented for core PG1205 from Basalt Lake in East Greenland are based on radiocarbon ages originally published by Wagner et al. (2000 in Palaeo3, 160: 45-68), although reference is made to the PhD thesis of Wagner (2000). The LANDO calculations suggest continuous and relatively constant sedimentation since at least 21 cal. ka BP. However, both Wagner et al. (2000) and Wagner (2000) state that the lake record consists of a till at its base, which in all likelyhood was deposited during the Milne Land stade 11.30 - 11.15 cal. ka BP, overlaid by ca. 6.4 m of glaciolacustrine sediments deposited with high sedimentation rates during deglaciation and ca. 2.6 m of hemipelagic sediments deposited with much lower rates during the past ca. 10 ka BP. Hence, the calculations conducted by Pfalz et al. obviously neglect the regional glacial history presented and discussed by Wagner et al. (2000) and Wagner (2000) as well as many papers published before and afterwards, giving the wrong impression that this part of East Greenland became deglaciated already prior to 21 cal. ka BP.*

> Agreed. To ensure that we do not include grey literature in the references, we now used Wagner et al., (2000) as reference for the core PG1205. Similar to the Lake Ladoga sediment core Co1309, we changed the length of the sediment core to the last dating point to avoid strong extrapolation in the new version.

*These two examples illustrate that neglecting geological evidence for hiatuses or large changes in the rates of deposition can create much larger errors in age-depth models and resulting sedimentation rates than the employment of an age-depth modelling system that may not be ideal for the record investigated. From the two examples it becomes evident to us that the literature existing for all sediment records used in the third case study, not only Co1309 and PG1205, needs to be (re)studied and discussed to assure that the geological evidence provided is considered in the sedimentation rates calculated.*

Thank you for bringing this important matter to our attention. We agree and have re-examined all the sediment records closely by comparing the originally published age-depth models and the LANDO outputs. We added an extra paragraph regarding the adjustments on the individual cores (Page 26, Lines 587-601).

---

## Referee Report (RR1)

LANDO Revised Paper
12th April 2022

The authors have done a very careful and good job of addressing the comments raised by the reviewers and community. I would like to thank them for considering our suggestions.

I have added a few (minor) suggestions to the pdf directly on grammatical edits where I felt slight rewording could ease understanding as an English speaker. The authors may wish to consider these but they are minor. The main ideas are clear already.

I did have two slightly more significant points explained below. The first regards a remaining (minor) uncertainty regarding reservoir ages. I may have misunderstood he text here. This could be resolved either way with the addition of the word "approximate" to the text.

The second is that I still feel there is a danger some users will see CS2 and think that LANDO will allow them to fit age-depth models to bad/incongrous data – and that it will automatically fix the underlying data disagreements. To me, this needs addressing much more clearly to prevent such a misunderstanding. I am not suggesting what they should be asked to do any further work – simply add a short statement preventing this possibility (and the removal of a line suggesting the LANDO ensemble will cover all options).

**Reservoir Ages**

I remain slightly confused by how they have worked out $^{14}C$ reservoir ages. Calculation of the reservoir age would require them to have a $^{14}C$ (marine/lake) sample (at the top of the core) for which they know the calendar age (e.g. 10 cal yr BP = AD 1940). The total reservoir age/hard-water effect is then the difference between the $^{14}C$ age of this (AD1940) sample and the IntCal20 estimate at 10 cal yr BP (AD 1940). For a $\Delta R$ we instead look at the offset to Marine20, i.e. the difference between the $^{14}C$ age of this sample and the Marine20 estimate at 10 cal yr BP. This is shown below. It is the vertical difference between $^{14}C$ ages for a fixed calendar age:

[Figure]

Now for a lake core, if we don't have a measurement at the top of the core (which you know/assume has a calendar age corresponding to the core drilling) you can't do this. You need a sample of known calendar age.

My understanding (??) from their text is that what they suggest instead is to:
a) Initially calibrate all their $^{14}$C dates assuming no reservoir effect against e.g. IntCal20. This provides (non-reservoir) calendar age estimates for each sample.
b) Fit a model through the resultant (non-reservoir) calendar ages and estimate the calendar age of the top of the core
c) Estimate the reservoir age as the difference between the calendar age at the top of the core and the year the core was drilled.

Have I understood this correctly? If so, this is effectively working out the horizontal difference (in cal years) between the sample and the curve in the figure above. This will give very slightly different values (as the IntCal curve is wiggly and not quite y = x)).

**Importantly,** if you do not have a $^{14}$C sample at the top of the core of known age, then you cannot do the first approach. Their proposed approach is therefore probably the best one can achieve as an approximation. I am not therefore suggesting they do anything differently (especially as they add an uncertainty).

**Suggested Action:** If my above understanding of their approach is correct, they might just wish to state this is an approximation (as it is all that it is possible), e.g., on line 164 just swap calculate for approximate (everything else can probably stay as it is)

We approximated the reservoir effect by subtracting the target age from the mean predicted age, whereas the associated error we based on the two-sigma uncertainty ranges of the prediction

If my understanding is not correct then they do not need to do anything (it may just be the fact I have not quire understood).

**Case Study 2 (CS2) – I feel much more strongly here that it is dangerous to suggest users can fit age-depth models to data like this. In my opinion, a much clearer/stronger caveat is needed**

Despite the author response, I still feel as though there is a significant danger that some readers will see case study CS2 and think LANDO will automatically and always solve issues of the divergent dates. I am still not sure that the LANDO ensemble covers all the potential options as it seems to have decided that the $^{14}$C dates around 400-500cm are all erroneous (no models extend up to here). Further BChron produces very bizarre sedimentation rates.

I understand the authors wish to use it as an example to show how the methods can provide very different age-depth models on pathological examples. That is fine so long as it doesn't give the impression to LANDO users that they can treat it as a perfect black-box. I fear the manuscript still gives an idea, especially lines 564 – 570, that LANDO and proxy optimisation can automatically fix datasets where the calendar ages are in huge disagreement.

The authors have added a caveat on lines 555 – 560, however I believe something stronger is needed here, with a similar statement in subsection 3.2 introducing this example. Placing the

caveat only in the discussion (on line 560) means many readers will miss it. This caveat should say that:

a) the first thing one should do in such an example of incongruent dates (and outliers) is to go back and use their expert knowledge to assess the reliability
b) use extreme caution fitting any age-depth model.
c) LANDO will not automatically fix the issue of divergent dates (even when the whole ensemble of 4/5 models is considered)

**Suggested Action 1:** Again, the authors need not redo anything but I would like to see something making the above points clear on pg17. For example, as the introductory paragraph:

**3.2. "Inconsistent sequence" – Case Study Number 2**

For the second case study we consider an example where the underlying calendar age estimates within the core are highly incongruous with one another, see Figure 3. Before considering modelling the age-depth relationship using data that disagree so strongly, a user must explore and aim to understand the reasons for any outliers. Fitting any age-depth model, including the LANDO ensemble, to such divergent data should be done with extreme caution and we do not recommend it. Here we primarily aim to illustrate the range of age-depth models obtained within the ensemble, and the process of optimisation using proxy-based lithography.

**Suggested Action 2:** I would also remove the paragraph from lines 564 – 570 which culd be read to suggest that the LANDO ensemble will always fix things. I do not see it is the case that a LANDO ensemble will include all possible age-depth relationships (both generally and for this specific example). One should really not be fitting automated models to this kind of data.

I do not think adding such caveats will affect the popularity/use of LANDO, since hopefully datasets as incongruous as CS2 are rare. It will however hopefully prevent bad inference based on bad data.

[revised manuscript text omitted]

---

## Author Response (AR2)

**Point-by-point changes and responses to comments**

**Referee Timothy J. Heaton**

Dear Timothy J. Heaton,

First of all, we would like to thank you very much for reviewing our manuscript for a second time. We appreciate your comments on improving the revised version of our manuscript.

We provide responses to each individual comment and our suggested changes and adjustments to the current manuscript below. We have highlighted your comments in black and *italics* and highlighted our responses in blue.

Thank you once again for taking the time to review our study.

On behalf of all the authors,

Gregor Pfalz
* * *
**Reservoir Ages**

*I remain slightly confused by how they have worked out 14C reservoir ages. Calculation of the reservoir age would require them to have a 14C (marine/lake) sample (at the top of the core) for which they know the calendar age (e.g. 10 cal yr BP = AD 1940). The total reservoir age/hard-water effect is then the difference between the 14C age of this (AD1940) sample and the IntCal20 estimate at 10 cal yr BP (AD 1940). For a $\Delta R$ we instead look at the offset to Marine20, i.e. the difference between the 14C age of this sample and the Marine20 estimate at 10 cal yr BP. This is shown below. It is the vertical difference between 14C ages for a fixed calendar age:*

[Figure]

*Now for a lake core, if we don't have a measurement at the top of the core (which you know/assume has a calendar age corresponding to the core drilling) you can't do this. You need a sample of known calendar age.*

*My understanding (??) from their text is that what they suggest instead is to:*
*a) Initially calibrate all their 14C dates assuming no reservoir effect against e.g. IntCal20. This provides (non-reservoir) calendar age estimates for each sample.*
*b) Fit a model through the resultant (non-reservoir) calendar ages and estimate the calendar age of the top of the core*
*c) Estimate the reservoir age as the difference between the calendar age at the top of the core and the year the core was drilled.*

*Have I understood this correctly? If so, this is effectively working out the horizontal difference (in cal years) between the sample and the curve in the figure above. This will give very slightly different values (as the IntCal curve is wiggly and not quite y = x)).*

***Importantly**, if you do not have a 14C sample at the top of the core of known age, then you cannot do the first approach. Their proposed approach is therefore probably the best one can achieve as an approximation. I am not therefore suggesting they do anything differently (especially as they add an uncertainty).*

***Suggested Action**: If my above understanding of their approach is correct, they might just wish to state this is an approximation (as it is all that it is possible), e.g., on line 164 just swap calculate for approximate (everything else can probably stay as it is)*

*We approximated the reservoir effect by subtracting the target age from the mean predicted age, whereas the associated error we based on the two-sigma uncertainty ranges of the prediction*

*If my understanding is not correct then they do not need to do anything (it may just be the fact I have not quire understood).*

> Thank you for your comment and suggestion. Yes, you understood our approach correctly. We agree with your suggestion and changed the sentence (Lines 163 to 164, Page 5) to: "We **approximated** the reservoir effect by subtracting the target age from the mean predicted age, whereas the associated error we based on the two-sigma uncertainty ranges of the prediction."

***Case Study 2 (CS2) – I feel much more strongly here that it is dangerous to suggest users can fit age-depth models to data like this. In my opinion, a much clearer/stronger caveat is needed***

*Despite the author response, I still feel as though there is a significant danger that some readers will see case study CS2 and think LANDO will automatically and always solve issues of the divergent dates. I am still not sure that the LANDO ensemble covers all the potential options as it*

*seems to have decided that the 14C dates around 400-500cm are all erroneous (no models extend up to here). Further BChron produces very bizarre sedimentation rates.*

*I understand the authors wish to use it as an example to show how the methods can provide very different age-depth models on pathological examples. That is fine so long as it doesn't give the impression to LANDO users that they can treat it as a perfect black-box. I fear the manuscript still gives an idea, especially lines 564 – 570, that LANDO and proxy optimisation can automatically fix datasets where the calendar ages are in huge disagreement.*

*The authors have added a caveat on lines 555 – 560, however I believe something stronger is needed here, with a similar statement in subsection 3.2 introducing this example. Placing the caveat only in the discussion (on line 560) means many readers will miss it. This caveat should say that:*
*a) the first thing one should do in such an example of incongruent dates (and outliers) is to go back and use their expert knowledge to assess the reliability*
*b) use extreme caution fitting any age-depth model.*
*c) LANDO will not automatically fix the issue of divergent dates (even when the whole ensemble of 4/5 models is considered)*

***Suggested Action 1***: *Again, the authors need not redo anything but I would like to see something making the above points clear on pg17. For example, as the introductory paragraph:*

*3.2. "Inconsistent sequence" – Case Study Number 2*
*For the second case study we consider an example where the underlying calendar age estimates within the core are highly incongruous with one another, see Figure 3. Before considering modelling the age-depth relationship using data that disagree so strongly, a user must explore and aim to understand the reasons for any outliers. Fitting any age-depth model, including the LANDO ensemble, to such divergent data should be done with extreme caution and we do not recommend it. Here we primarily aim to illustrate the range of age-depth models obtained within the ensemble, and the process of optimisation using proxy-based lithography.*

***Suggested Action 2***: *I would also remove the paragraph from lines 564 – 570 which could be read to suggest that the LANDO ensemble will always fix things. I do not see it is the case that a LANDO ensemble will include all possible age-depth relationships (both generally and for this specific example). One should really not be fitting automated models to this kind of data.*

*I do not think adding such caveats will affect the popularity/use of LANDO, since hopefully datasets as incongruous as CS2 are rare. It will however hopefully prevent bad inference based on bad data.*

Thank you for the detailed comment and your concerns. As you rightly pointed out, we wanted to show CS2 as an example, but not as the ultimate solution to scattered data. We agree with your first suggested action and have added the following paragraph on page 17: *"For the second case study, we considered an example where the underlying age determination data within the core are very contradictory to each other (see Figure 3). Before considering modeling such an age-depth relationship with conflicting data, users*

*need to investigate and try to understand the reasons for any outliers. Fitting any age-depth model, including the LANDO ensemble, to such divergent data should be done with extreme caution and we do not recommend doing so without further deliberate investigation. Here we primarily aim to illustrate the range of age-depth models obtained within the ensemble as well as the results of the optimization with our proxy-based lithology."*

Regarding the second proposed action, we removed most of the paragraph, except the last sentence, which says:

*"For palaeoenvironmental reconstruction, users should also propagate these increased uncertainties into their proxy interpretation, which is often underrepresented (Lacourse and Gajewski, 2020; McKay et al., 2021)."*

We added this sentence to the previous paragraph, because with this statement we want to ensure that users account for age-depth model uncertainty in their further analysis, regardless of which models they use.

Line 40, Page 1: Change "Lakes" to "Lake (singular)

Changed.

Line 71, Page 2: Change "Implemented methods" to "Methods have been implemented"

Changed.

Line 72, Page 2: Add "to" before "support"

Added.

Line 80, Page 2: Change "usually, users only select" to "user usually only select"

Changed.

Line 149, Page 5, Table 1: Should this be "or" rather than "and" to correspond with age above?

Correct - we changed it to "or".

Line 174, Page 6: Change "create" to "creating"

Changed.

Lines 185 to 186, Page 6: Split sentence into two sentences to "… modeling outcome. This is challenging if no objective prior knowledge exists"

Agreed. We separated the sentence into two sentences: *"As mentioned before, the selection of model priors and parameters has an impact on the* **modeling outcome. This is challenging if no objective prior knowledge exist.**"

Lines 192 to 193, Page 6: Change to "…for problematic cores, as recommended by Blaauw et al. (2021)." so it agrees with author response

    Changed.

Line 206, Page 7: Add "an" before "attribute"

    Added.

Line 238, Page 8: Change commas to "As a counterexample, for the second study we have chosen"

    We changed the sentence to: *"**As a counterexample, for the second case study we have chosen** the sediment core EN18208 (Vyse et al., 2020)."*

Line 293, Page 13: Change "extract" to "extracted"

    Changed.

Line 535, Page 25: Add "a" before "significant"

    Added.

Line 664, Page 28: Change to "provides neither a query optimizer, nor relies on Map-Shuffle-Reduce"

    We changed the sentence to: *"The key difference is that "Dask" **provides neither a query optimizer, nor relies on Map-Shuffle-Reduce,** a data processing technique for distributed computing, but instead uses a generic task scheduling (cf. Dask Development Team, 2016)."*

Line 684, Page 29: Change "result" to "resulting"

    Changed.

Line 696, Page 29: Change order to "can increase this factor even further by…"

    We changed this sentence to: *"When considering that our setup consisted of six CPUs (12 threads) and 16 GB RAM, user **can increase this factor even further by** using larger computing facilities."*

Line 705, Page 29: Change "bares" to "bears"

    Changed.

Line 713, Page 29: Remove "do not" and "more"

Removed.

Line 726, Page 30: Change "most" to "best"

Changed.

Line 733, Page 30: Change "most" to "better"

Changed.